# A Hitchhiker's Guide to Fine-Grained Face Forgery Detection Using Common Sense Reasoning

**Niki Maria Foteinopoulou**[1]    **Enjie Ghorbel**[1,2]    **Djamila Aouada**[1]

[1]CVI[2], SnT, University of Luxembourg

[2]Cristal Laboratory, National School of Computer Sciences, University of Manouba

{niki.foteinopoulou, enjie.ghorbel, djamila.aouada}@uni.lu

## Abstract

Explainability in artificial intelligence is crucial for restoring trust, particularly in areas like face forgery detection, where viewers often struggle to distinguish between real and fabricated content. Vision and Large Language Models (VLLM) bridge computer vision and natural language, offering numerous applications driven by strong common-sense reasoning. Despite their success in various tasks, the potential of vision and language remains underexplored in face forgery detection, where they hold promise for enhancing explainability by leveraging the intrinsic reasoning capabilities of language to analyse fine-grained manipulation areas. For that reason, few works have recently started to frame the problem of deepfake detection as a Visual Question Answering (VQA) task, nevertheless omitting the realistic and informative multi-label setting. With the rapid advances in the field of VLLM, an exponential rise of investigations in that direction is expected. As such, there is a need for a clear experimental methodology that converts face forgery detection to a Visual Question Answering (VQA) task to systematically and fairly evaluate different VLLM architectures. Previous evaluation studies in deepfake detection have mostly focused on the simpler binary task, overlooking evaluation protocols for multi-label fine-grained detection and text-generative models. We propose a multi-staged approach that diverges from the traditional binary evaluation protocol and conducts a comprehensive evaluation study to compare the capabilities of several VLLMs in this context. In the first stage, we assess the models' performance on the binary task and their sensitivity to given instructions using several prompts. In the second stage, we delve deeper into fine-grained detection by identifying areas of manipulation in a multiple-choice VQA setting. In the third stage, we convert the fine-grained detection to an open-ended question and compare several matching strategies for the multi-label classification task. Finally, we qualitatively evaluate the fine-grained responses of the VLLMs included in the benchmark. We apply our benchmark to several popular models, providing a detailed comparison of binary, multiple-choice, and open-ended VQA evaluation across seven datasets. https://nickyfot.github.io/hitchhickersguide.github.io/

## 1 Introduction

Recent developments in deep generative modelling have resulted in hyper-realistic synthetic images/videos with no clear visible artefacts, making the viewers question whether they can still trust their eyes. Unfortunately, despite its relevance in a wide range of applications, such technology poses a threat to society as it can be used for malicious activities [16]. In a world where synthetic images of a person, known as deepfakes, can easily be generated, it becomes crucial to fight misin-

38th Conference on Neural Information Processing Systems (NeurIPS 2024) Track on Datasets and Benchmarks.

formation not only by identifying manipulated images/videos in an automated manner but also by explaining the decision behind this classification to reinstate trust in Artificial Intelligence.

Numerous successful works on deepfake detection have been proposed in the literature to tackle the risks of face forgery [37, 77, 52, 7, 81]. Existing methods primarily rely on deep binary classifiers, resulting in black-box models that predict whether an input is real or fake. Consequently, explaining why those predictions are being made is not straightforward. To handle this issue, a few interpretable deepfake detection methods have been introduced by examining attention maps or weight activations [88, 72] to identify fine-grained areas of manipulation; however, these are based on a post-hoc analysis and thereby do not intrinsically incorporate an explainable mechanism. On the other hand, Vision Large Language Models (VLLMs) have emerged as a pioneering branch of generative Artificial Intelligence (AI), showcasing advancements in common sense reasoning and an inherent explainability that arises from the intrinsic nature of language [22]. They have demonstrated impressive capabilities in tasks such as Visual Question Answering (VQA) [47] and the generation of descriptive content for downstream applications [71], hence bridging the gap between vision-language understanding and contextual reasoning. However, despite these achievements, the explainable power of VLLMs remains under-explored in the field of deepfake detection, with only a handful of works mostly exploring the vision and language capabilities for the binary classification of fake/real images [26, 8, 31, 70] all of which are evaluated on different benchmarks and metrics. To the best of our knowledge, Zhang *et al.* [87] is the only work employing a VQA approach in deepfake detection by proposing to extend the FF++ dataset with captions in natural language generated by humans. However, this work targets only one manipulated region at once, while deepfakes can incorporate several stacked manipulations [63, 68]. Moreover, the provided augmented FF++ dataset does not allow for cross-dataset evaluation in a VQA setting without an extensive annotation effort, making it difficult to investigate the generalisation aspect. In addition, the fine-grained evaluation in previous works is limited as the more challenging open-ended VQA task is not explored.

Explainable fine-grained detection –that is, identifying manipulation beyond the binary fake/real decision– in natural language is still in its infancy. However, as VLLM works for deepfake detection are expected to appear following the overwhelming success of foundation models in other tasks, two research questions need to be addressed: 1) *"To what extent can existing VLLMs detect deepfake images and what rationale supports the decision?"* and 2) *"How do we fairly and comprehensively evaluate VLLMs in the fine-grained task?"*. In deepfake detection, benchmarks have mainly focused on binary or multi-class decisions and discriminative networks [82, 87], making them unsuitable to answer these research questions. Indeed, they do not propose a unified method to match the generated responses to fine-grained multi-label categories. Similarly, existing benchmarks in Visual Question Answering (VQA) [19, 10] primarily address multi-class tasks, which may not be suitable for the multi-label nature of fine-grained deepfake detection as highlighted in [63].

In this work, our objective is to conduct a thorough quantitative and qualitative evaluation of VLLMs for the task of fine-grained multi-label deepfake detection in a systematic and scientific approach, employing a multi-stage protocol without costly human captioning efforts. In the first stage, we assess the models' performance on the binary task using various prompts while also evaluating the model's sensitivity to the provided instructions. In the second stage, we delve into multi-label fine-grained detection, aiming to identify areas of manipulation within a multiple-choice Visual Question Answering (VQA) framework, i.e. what areas from a given list are identified as manipulated. Subsequently, in the third stage, we extend our investigation by converting the fine-grained detection task into an open-ended question –that is, identifying areas without instructing the model to select from a list of categories. Here, as the task is a multi-label problem, we compare two matching strategies: a) using the cosine similarity between the generated text and ground truth labels and b) counting the occurrence of the class name in the generated text. Finally, we qualitatively evaluate the fine-grained responses generated by the VLLMs included in our benchmark, providing nuanced and new insights into their performance.

The main contributions of this work can be thus summarised as follows:

- We introduce a novel evaluation protocol for deepfake detection under the Visual Question Answering (VQA) **multi-label setting** and without the use of human annotations. This is different from [87] that is based on a succession of yes/no questions for fine-grained areas, resulting in a binary classification setting and relies on a relatively small dataset which cannot be extended without costly annotation efforts. In addition, our multi-stage protocol

allows for open-ended VQA evaluation, which is a more challenging task. To the best of our knowledge, this is the first work to do so in the field of deepfake detection, offering a fresh perspective on explainability through fine-grained multi-label analysis.

- We present a systematic, unified evaluation study of current state-of-the-art (SOTA) VLLMs, facilitating consistent assessment across different models. This framework is designed to be extendable to any existing or future deepfake dataset, ensuring fair and comprehensive comparisons with future models, thus promoting transparency and reproducibility in the evaluation process.

- Through extensive comparison and analysis of the tested models and an ensemble of models, our study yields new insights into the capabilities and limitations of VLLMs in the context of deepfake detection. We will use these insights to advance research in the domain and hope to inform future developments and optimisations in model design and evaluation strategies.

## 2 Related Work

**DeepFake generation** encompasses various forms of facial manipulation, including face reenactment [6, 1], face swapping [24, 45], and entirely generated face images [32, 69]. **Deepfake detection** algorithms classify samples as real or fake [7, 81, 77, 37], relying on artefacts left by manipulation methods, often analysed qualitatively for explainability [21, 91]. However, this qualitative analysis happens on a secondary stage and primarily depends on human observers. While generative methods often use natural language instructions [54, 75, 58, 55, 78], explaining manipulations in natural language –a natural extension of the generation process to detection– is still an emerging area.

With the rise of VLLMs, recent works [8, 31, 70, 30, 73] explore vision and language for face forgery detection, primarily focusing on binary detection in a retrieval setting, with fewer [70, 52, 91] examining fine-grained areas usually as a secondary task. The latter have relied on generated pseudo-fake datasets to improve generalisation [70, 52, 91], which have a major drawback –that is, the use of pseudo-fake datasets hampers fair comparisons and does not reflect the current state-of-the-art in deepfake generation.

Several **VLLMs** foundation models [4, 15, 35, 44, 43], bridge the gap between vision and language. These are typically trained on very large datasets with general knowledge. As the computational and data resources needed to train VLLMs from scratch are very high, numerous works leverage the pre-trained networks in three main directions: a) exploring the latent feature space [14, 57] of vision and language, b) parameter efficient training in a downstream task [39, 74, 18] and c) evaluating foundation models in new domains [65, 76].

**Benchmarks** for classification tasks [19, 38, 2] in a VQA setting typically address the multi-class paradigm, which may not be appropriate for addressing explainability in DeepFake detection by adopting a multi-label fine-grained strategy, as several areas can be manipulated at once. A few preliminary works in DeepFake detection [3, 26, 87] benchmark ChatGPT4[1] and Gemini[2]; however, these have primarily focused on the more straightforward binary task and did not explore the reasoning capabilities of VLLMs for fine-grained labels. Furthermore, both these works focused on VLLMs that are not open-sourced with limited information regarding their training set and architecture; thus, it is not possible to assess whether the benchmarks are, in fact, zero-shot or whether they have been trained on deepfake-related image-language pairs. Zhang *et al.* [87] propose extending FF++ annotations with captions in natural language using human annotators. However, this method is limited to binary decisions, while a given deepfake image can undergo several manipulations. Furthermore, it does not explore the open-ended VQA setting and does not offer a method for cross-dataset evaluation without a costly annotation process. Within DeepFake detection, the vast majority of benchmarking works [60, 65, 46, 82, 34] have focused on binary discriminative networks and are therefore not fit to evaluate the capabilities of generative models such as VLLM, particularly for fine-grained labels.

In a nutshell, the main novelty of this benchmark compared to previous works [19, 82, 38, 2, 60, 65, 46, 34] is threefold: 1) it converts the multi-label classification task of face forgery detection to a

---

[1]https://openai.com/gpt-4
[2]https://gemini.google.com

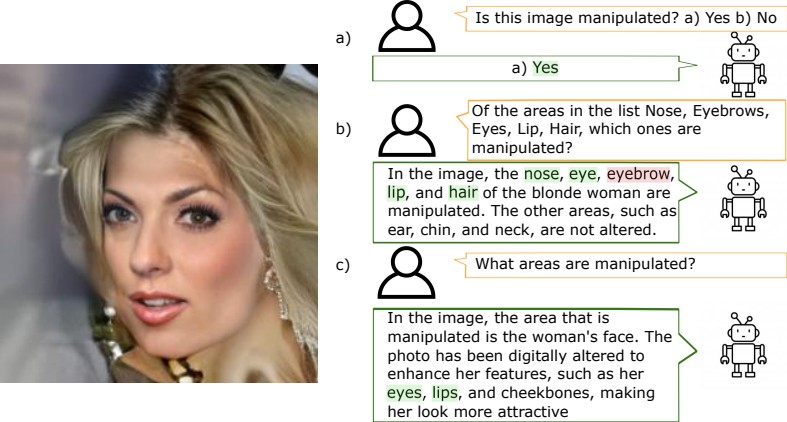

Figure 1: Overview of the proposed benchmarking method, using multiple stages to evaluate the performance of VLLMs in the context of deepfake detection. In the first stage (a), we assess the binary classification performance of VLLMs. In the second stage (b), we perform a fine-grained classification using multiple-choice instruction. In the third and final stage (c), we ask the model to identify fine-grained areas in open-ended VQA. The image example [3]is a sample from the SeqDeepFake dataset [63], and responses are generated using Llava-1.5 [43]

VQA task so that VLLM's common sense reasoning capabilities can be evaluated, 2) it systematically and consistently assesses VLLM capabilities on nine binary and three fine-grained benchmarks and 3) is offering an open source and extendable framework for future zero-shot or task-specific VLLMs, that ensures a fair comparison.

## 3 Common Sense Reasoning for Face Forgery Detection

**Preliminaries:** We formalise the language generation process of VLLM architectures, akin to standard VQA models, where the model is prompted with an image and a query to produce an autoregressive answer. Given an image $\mathbf{X}_v \in \mathbb{R}^{H \times W \times C}$ and a text prompt $\mathbf{X}_t \in \mathbb{R}^{L \times d}$ as input, a sentence $\psi$ is generated represented as a sequence of word tokens. The generation can be represented by the function $p(\psi|\mathbf{X}_v, \mathbf{X}_t) = \prod_{j=0}^{|\psi|} p(\psi_j|\psi_{<j}, \mathbf{X}_v, \mathbf{X}_t)$, where $H \times W \times C$ represent the image dimension, $L$ is the number of tokens, $d$ is the embedding dimension, $\psi = (\psi)_{0 \leq j < |\psi|}$ is the generated sentence, and $|.|$ the cardinality. In VQA tasks, the model response aims to match human annotations. This task differs from typical classification problems due to the diverse semantic nature of questions and answers in natural language. The evaluation protocol is outlined for the binary case in Section 3.1 and for open-ended evaluation and multiple-choice of fine-grained labels in Section 3.2. An overview of the proposed method is shown in Figure 1.

### 3.1 Binary Classification to VQA

In binary classification, the task is to predict whether the image sample is a product of face forgery. We create a benchmark to assess VLLM capability in binary Deepfake detection by transforming the discriminative task into a VQA problem. We consider only the positive category for each image $\mathbf{X}_v$ to generate the relevant instruction; that is, we limit the prompt to identifying whether an image is a Deepfake and not whether it is a genuine sample. The prompt used is in the form:

$$\mathbf{X}_t = \text{``Is this image } [s_i] \text{ ? a) Yes b) No''} \tag{1}$$

where $\mathbf{s}_i \in \mathcal{S}$ denotes a set of standard terms used to describe deep fakes in the English language. The synonyms are employed to assess the reasoning capabilities of each tested model by investigating whether the understanding of the model is robust to the given instruction.

---

[3]Ground truth: Hair, Nose, Lip, Eye

## 3.2 Fine-Grained Labels:

Fine-grained labels typically refer to manipulation areas. Predicting them necessitates the use of multi-label classification, as multiple areas can be manipulated at once. Following the initial binary prompt, a follow-up prompt to explain what areas of manipulation are identified is given to the VLLM with the same image, as shown in Fig. 1. We propose using two versions of follow-up prompts, one as an open-ended question and one as a multiple-choice. Specifically, the open-ended follow-up prompt follows the template:

$$\mathbf{X}_t = \text{``What area of this image is } [\mathbf{s}_i] \text{ ?''} \tag{2}$$

For the multiple-choice instruction, we follow the template:

$$\mathbf{X}_t = \text{``Of the areas in the list } [\mathbf{cls}_0, \ldots, \mathbf{cls}_{|\mathcal{C}|}], \text{ which ones are } [\mathbf{s}_i] \text{ ?''} \tag{3}$$

where $\mathbf{cls}_i \in \mathcal{C}$ is the class name of the $i$-th class from the set of target classes $\mathcal{C}$.

## 3.3 Matching Strategies:

To evaluate the generated responses, we employ several matching methods depending on the task. The stricter one uses an *Exact Match (EM)* approach, that estimates whether the generated sentence $\psi$ is exactly equal to the class name $cls_i$:

$$p(\hat{y}_i) = \begin{cases} 1 & \text{if } \psi \equiv \mathbf{cls}_i \\ 0 & \text{if } \psi \neq \mathbf{cls}_i \end{cases} \tag{4}$$

where $\hat{y}_i$ is the prediction for the $i$-th class. In the given task, an answer is considered correct only if the model output exactly matches the class names or 'Yes'/'No' in the binary case. As the responses tend to be longer for fine-grained classification and reflect reasoning in natural language for a multi-label problem, a more appropriate matching strategy is to consider a response correct if the class name is *Contained* in the response, as proposed by Xu *et al.* [76] – that is $p(\hat{y}_i) = 1$, if $\mathbf{cls}_i \in \psi$ and $0$ otherwise. We extend this to include synonyms of class names, as several ways exist to describe some areas (e.g. "Bangs" could also be described as "Hair"). Finally, we propose adapting the text-to-text score (*CLIP distance*) proposed by Conti *et al.* [14] for the multi-label task. This is done by using a sigmoid function over the cosine similarity matrix of the prediction embeddings and class name embeddings (obtained with a CLIP [59] text encoder), using an empirical temperature $t$ of 0.5 so that $p(\hat{y}_i) = \sigma(<\psi, \mathbf{cls}_i > \frac{1}{t})$. The symbol $< ., . >$ denotes the cosine similarity of the text embeddings and $\sigma(.)$ is the sigmoid function.

## 4 Experimental Set-Up

### 4.1 Tested VLLMs

We select four open-source state-of-the-art VLLMs to be included in this benchmark; specifically, we include LlaVa-1.5 [43] (an improved version of the LlaVa architecture [44]), BLIP2 [35] and finally InstructBlip [15] with Flan-T5 and Flan-T5-xxl language generators [13]. Finally, for the binary task, we include the CLIP [59] performance as a baseline and compare it against an ensemble of BLIP2 [35] and LlaVa-1.5 [43] following the ensembling strategies for VQA tasks [5], and GPT4v as an upper bound[4]. Experiments using GPT4v are performed on a subset of 5k samples selected from each dataset, and thus, the results may be susceptible to some degree of bias from the sampling, which needs to be considered when comparing the models. The selection of the VLLM is guided by three factors. First, we select architectures with publicly available weights and training methods to ensure transparency and fairness in the evaluation. Second, we include methods that generate output in Natural Language rather than a set of features or classification predictions. Finally, we select methods that have achieved state-of-the-art performance on several zero-shot tasks. Additional model details, such as the number of parameters and pre-training information of the tested models, can be found in Appendix A.

---

[4]https://openai.com/index/gpt-4v-system-card/

## 4.2 Datasets

We evaluate the performance of our method on seven published challenging benchmarks and one pseudo-fake dataset; more specifically, seven datasets for binary detection and two for the fine-grained task. All are evaluated at the frame level, as in previous image works [80, 52, 49]. **FF++:** [60] consists of over 20k images of DeepFake images from 1000 videos, using four types of manipulation methods: Deepfakes, Face2Face, FaceSwap and NeuralTextures. The dataset is split into train, validation, and test with an 80%, 10%, and 10% split, respectively. **DFDC:** [17] is composed of 5k videos of real and manipulated faces split into 4,464 unique training clips and 780 unique test clips. **Celeb-DF:** [41] includes 590 original videos collected from YouTube with subjects of different ages, ethnic groups and genders, and 5639 corresponding manipulated videos. **WildDeep-Fake:** [90] is a challenging dataset for in-the-wild detection, which consists of 7,314 face sequences extracted from 707 videos that are collected completely from the internet. **StyleGAN:** Two sub-sets consist of curated images generated with StyleGAN3 [29] and StyleGAN2 [28] along with original ones for binary detection of facial images. **SeqDeepFake:** [63] dataset consists of 85k sequential manipulated face images based on two representative facial manipulation techniques, facial components manipulation [32] and facial attributes manipulation [27]. The labels include annotations of manipulation sequences with diverse sequence lengths. **R-splicer:** Augmenting real data by generating pseudo-fake images is a common practice in deepfake detection [36, 49, 11, 66, 89, 40]. Such methods simulate characteristic face-swap artefacts using simplistic operations on a predefined set of regions. In this work, we use a spliced dataset of 59k images to evaluate fine-grained labels of five regions –entire face, mouth, nose, eyes, eyebrows– as implemented by Mejri *et al.* [49].

## 4.3 Metrics

**Accuracy** and the Area Under the Receiver Operating Characteristics (ROC) Curve (**AUC**) are the most common metrics used in DeepFake detection [82, 52, 65]. However, as the datasets in the task are massively imbalanced, we also use the harmonic mean of Precision and Recall (**F1-score**) for the binary task. Furthermore, we note that as AUC is a measure of the classifier's performance at different thresholds, it has very limited value in the VQA task where matching strategies result in polarised decisions; however, we include it for reference. In the fine-grained task, we use mean Average Precision (**mAP**), AUC and F1-score as indicators of classification performance.

## 5 Results

### 5.1 Binary Classification

**Robustness to different prompts:** We use seven synonyms for the positive class: "manipulated", "deepfake", "synthetic", "altered", "fabricated", "face forgery" and "falsified". As the binary task is simple and the instruction format is a 'Yes' or 'No' question, we use *EM* as a matching approach in this evaluation. In Fig. 2, we see the performance of each tested model under the binary detection setting on the two sub-sets of SeqDeepFake [63] and the R-splicer dataset using the three best performing synonyms: "manipulated", "synthetic" and "altered". The first observation is that no VLLM clearly outperforms others across all datasets and metrics. However, we see that BLIP-2 [35] has the most robust performance to the given instruction, even though it is the smallest in terms of parameters. Furthermore, the additional parameters of T5-xxl [13] do not seem to aid the task compared to the base InstructBLIP [15] with T5 generator, as the base model performs comparably better across most benchmarks. We theorise that as the VLLMs have not been explicitly trained on image-language pairs of manipulated images, a large number of parameters on the language generation leads to more hallucinations [25, 79] for this simple but abstract task. Compared to the CLIP [59], models appear to have competitive performance with the exception of InstructBLIP with T5xxl LLM. When both base models, i.e. BLIP-2 [35] and LlaVa [43], have relatively good performance, the ensemble shows marginal improvement, particularly in terms of Accuracy and F1; however, this is not consistent therefore we do not continue the investigation to fine-grained labels. The detailed performance of all models and synonyms across all datasets and additional analysis on CLIP [59] features can be found in the Appendix.

**Overall model performance:** We average the performance of the three best-performing synonyms on all nine benchmarks in Fig. 3. No model clearly outperforms others across all metrics and

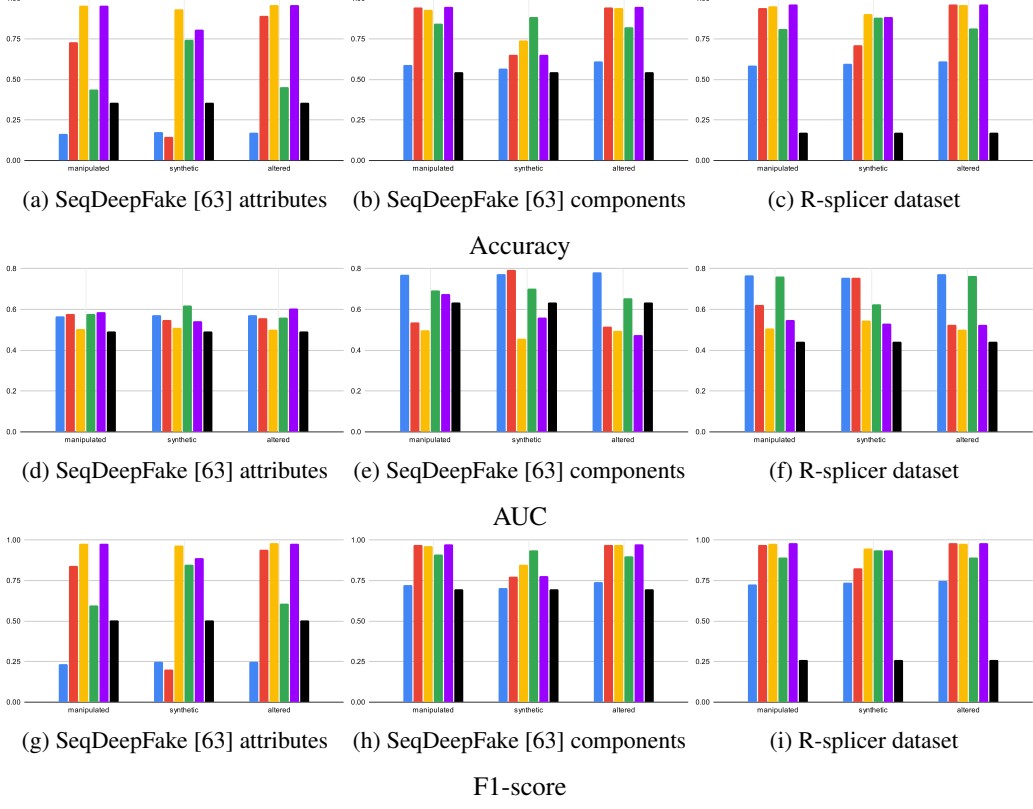

Figure 2: Exact Match *(EM)* Performance of each VLLM in terms of Accuracy (top), AUC (mid) and F1-score (bottom) for the top 3 synonyms "manipulated", "synthetic" and "altered"

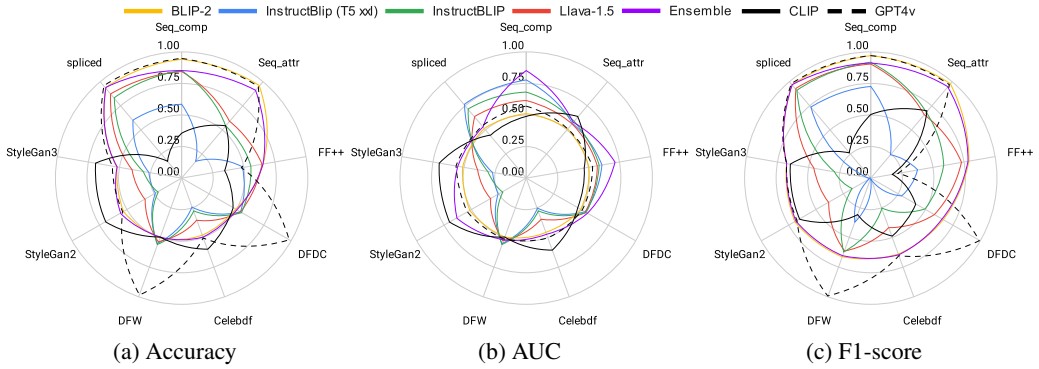

Figure 3: Exact Match *(EM)* Performance of each VLLM on all nine benchmarks

datasets; however, we can observe competitive performance from BLIP-2 [35] on the binary task, even though it is the smallest model in terms of parameters. We also see that all models struggle with the more challenging in-the-wild datasets, such as CelebDF [41], which highlights the need for further development to achieve adequate generalisation. Performance of GPT4v should be treated as an upper bound as we cannot assess whether the model has been trained on samples of the selected datasets. We see that GPT4v vastly outperforms the selected VLLMs on three benchmarks and has comparable performance on the rest, with the exception of FF++.

**Vision Encoder Finetuning:** We finetune contrastively the vision encoder of LlaVa on FF++ using a sigmoid loss [83] over an ensemble of prompts for the real/fake categories, and evaluate as described in the previous section. Training details for the vision encoder can be found in Appendix C. The

architecture with the fine-tuned vision encoder shows improved within dataset and cross-dataset performance as shown in Tab. 1. Even without detailed captions or updating the LLM weights, we see there are still gains from a task specific vision encoder, particularly in terms of F1-score with an average improvement of nearly $4\%$ within dataset and nearly $2\%$ cross-dataset (for SeqDeepFake, CelebDF and StyleGAN2).

Table 1: Binary performance of LlaVa-1.5 [43] with a fine-tuned Vision Encoder against the zero-shot baseline.

| | LlaVa Baseline | | LlaVa w. fine-tuned Vision Encoder | |
| --- | --- | --- | --- | --- |
| | Acc. | F1 | Acc. | F1 |
| FF++ | 64.54 | 73.10 | 64.57 (+0.03%) | 76.83 (+3.73%) |
| SeqDeepfake Attr. | 58.92 | 66.15 | 61.22 (+2.30%) | 68.03 (+1.88%) |
| SeqDeepfake Comp. | 84.62 | 90.57 | 84.24 (-0.37%) | 90.20 (-0.38%) |
| R-Splicer | 87.11 | 92.50 | 87.31 (+0.20%) | 92.62 (+0.12%) |
| DFDC | 54.02 | 58.24 | 53.86 (-0.16%) | 58.65 (+0.41%) |
| CelebDF | 35.67 | 41.81 | 37.60 (+1.93%) | 43.53 (+1.72%) |
| DFW | 53.35 | 61.56 | 53.45 (+0.10%) | 61.90 (+0.34%) |
| StyleGAN2 | 33.67 | 38.62 | 35.20 (+1.53%) | 39.72 (+1.10%) |
| StyleGAN3 | 39.80 | 45.66 | 39.30 (-0.50%) | 45.74 (+0.08%) |

**Metrics:** In terms of the selected metrics, following the initial intuition, there is limited information we can get from the standard Accuracy and AUC used in the binary task. Both are heavily skewed by the label distribution, which is typically imbalanced in deepfake datasets; however, the latter may also not be fit for VLLMs as AUC measures performance at different thresholds, which are not present with *EM* and *contains* matching strategies. As such, we argue that for the task at hand, the F1-score –and consequently robust to imbalance metrics– are more appropriate.

## 5.2 Finegrained Evaluation

For the fine-grained task, we evaluate the performance of the selected models in the open and closed vocabulary settings. The fine-grained labels are evaluated on samples where the ground truth is positive – i.e., on DeepFake samples.

**Open-Ended VQA:** We first evaluate the selected VLLMs under the open vocabulary VQA setting on the three fine-grained datasets. The results using *contains* and *CLIP distance* matching are reported in Tab. 2a and Tab. 2b respectively. An *EM* strategy is not possible in multi-label tasks, so no such evaluation is performed. No model clearly outperforms others across all metrics and datasets. In fact, we can observe that, in most cases, they have comparable performance. This holds true for both *contains* and *CLIP distance* metrics. In terms of matching strategy, using the *CLIP distance* consistently and greatly improves recall, as is evident by the improvement in the F1-score and explicitly shown in Appendix H. This matching approach slightly lowers the mAP and AUC scores compared to the *contains* metrics; however, using the cosine distance to match the open-ended responses to the class categories semantically may offer a more reliable output for the class of interest, as seen by the F1-score.

**Multiple choice VQA:** The performance of the VLLMs on the multiple-choice instruction is shown in Fig. 4. Even though the open-ended setting is theoretically more challenging, the performances of all tested models are comparable to each other and worse on the multiple-choice instruction for both mAP and AUC. Regarding the F1-score, however, LlaVa [43] consistently performs better than other models. Under the multiple-choice setting, we observe that the models tend to mention all label names, which raises the number of False Positives –a significant limitation of the multiple choice setting– or respond with "All of them" or "None of them", which makes matching of any sort more challenging and is reflected even more in the lower F1 score. Appendix G includes detailed metrics for each category.

Table 2: Model performance on open-ended fine-grained detection using a) *contains* and b *CLIP* matching

| | BLIP-2 | | | InstructBLIP | | | InstructBLIP-xxl | | | LlaVa-1.5 | | |
|---|---|---|---|---|---|---|---|---|---|---|---|---|
| | mAP | AUC | F1 | mAP | AUC | F1 | mAP | AUC | F1 | mAP | AUC | F1 |
| SeqDeepFake [63] attributes | 61.8 | 51.0 | 20.4 | 61.3 | 50.4 | 18.3 | **63.1** | **53.6** | _37.5_ | _61.7_ | _51.1_ | **40.0** |
| SeqDeepFake [63] components | 59.5 | 50.5 | 14.7 | _59.2_ | _50.0_ | 4.1 | **60.2** | **51.8** | **26.2** | 59.0 | 49.6 | _17.1_ |
| R-Splicer | 55.8 | 55.6 | 31.3 | 52.3 | 53.2 | 23.5 | _53.8_ | _54.0_ | _31.1_ | **58.7** | **57.5** | **41.6** |

(a) Assessment of model performance during open-ended evaluation with *contains* distance matching.

| | BLIP-2 | | | InstructBLIP | | | InstructBLIP-xxl | | | LlaVa-1.5 | | |
|---|---|---|---|---|---|---|---|---|---|---|---|---|
| | mAP | AUC | F1 | mAP | AUC | F1 | mAP | AUC | F1 | mAP | AUC | F1 |
| SeqDeepFake [63] attributes | **63.0** | **53.6** | 73.5 | 59.9 | 50.9 | _74.0_ | 60.4 | 50.7 | 55.5 | _61.0_ | _51.3_ | **74.1** |
| SeqDeepFake [63] components | _58.8_ | _52.7_ | _71.0_ | 55.5 | 49.0 | 71.7 | **59.9** | **55.7** | 59.8 | 56.1 | 49.6 | **71.7** |
| R-Splicer | _54.3_ | _55.3_ | 66.2 | 48.5 | 49.3 | _66.5_ | 54.0 | 53.1 | 60.3 | **56.7** | **57.4** | 66.5 |

(b) Assessment of model performance during open-ended evaluation with *CLIP* distance matching.

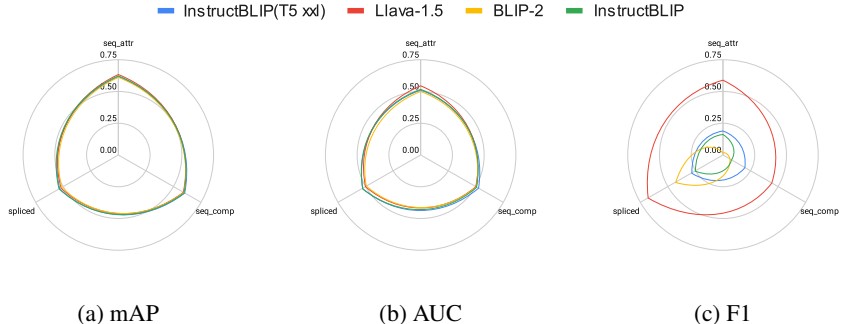

(a) mAP  (b) AUC  (c) F1

Figure 4: Assessment of model performance in multiple-choice settings, in terms of a) mAP, b) AUC and c) F1 during multiple-choice evaluation with *contains* matching.

## 5.3 Qualitative Evaluation

As the BertScore [86] is shown to correlate with human evaluation, we first present the Bertscore precision, recall, and F1 scores achieved by each model for the fine-grained open-ended responses compared with ground truth references that have been formatted using the prompt: "The areas that are $[\mathbf{s}_i]$ are $[\mathbf{c}ls_i]$". The results of this evaluation, along with the score of human annotators [19, 10] on a subset of the R-Splicer dataset, are shown in Tab. 3. As in previous sections, no model clearly outperforms others across all benchmarks; however, we see that Llava-1.5 [43] has the most competitive performance for most benchmarks, closely followed by InstructBLIP [15]. This is consistent with qualitative evaluations on VQA tasks [19, 44].

**Overall performance:** In the simpler binary setting, BLIP-2 is more robust to instruction than other models with more parameters; however, when it comes to fine-grained evaluation, larger models show an advantage in reasoning and identifying areas of manipulation in the open-ended and multiple-choice settings. It is, however, worth noting that no model clearly outperforms others across all metrics and datasets. All of the results presented are based on zero-shot evaluations, where models are tested without being specifically trained for deepfake detection. Despite this, the models are able to leverage a semantic mapping between language and visual input from their very vast pre-training, giving them an inherent concept of "real" versus "fake." This capability suggests that these models possess some degree of understanding when it comes to identifying deepfakes. However, this general understanding falls far behind that of task-specific models. When we fine-tune the vision encoder, there is a notable improvement in performance. The vision-language models can

Table 3: Open-ended qualitative evaluation with human annotators in Tab. a and BertScore [86] in Tab. b- d

| Model | Human Eval. Score |
|---|---|
| BLIP-2 | 0.35 |
| InstructBLIP | 0.36 |
| InstructBLIP-xxl | 0.33 |
| LlaVa-1.5 | **0.38** |

(a) Average score of Human Evaluation (R-splicer)

| Model | Precision | Recall | F1 |
|---|---|---|---|
| BLIP-2 | 79.77 | 78.75 | 79.24 |
| InstructBLIP | **86.53** | 83.22 | 84.81 |
| InstructBLIP-xxl | 80.73 | 81.78 | 81.25 |
| LlaVa-1.5 | 84.86 | **85.31** | **85.08** |

(b) SeqDeepFake [63] attributes

| Model | Precision | Recall | F1 |
|---|---|---|---|
| BLIP-2 | 79.87 | 79.72 | 79.61 |
| InstructBLIP | 81.12 | 83.89 | **86.90** |
| InstructBLIP-xxl | 82.57 | 81.77 | 81.01 |
| LlaVa-1.5 | **87.40** | **86.37** | 85.39 |

(c) SeqDeepFake [63] components

| Model | Precision | Recall | F1 |
|---|---|---|---|
| BLIP-2 | 79.55 | 79.76 | 80.04 |
| InstructBLIP | 83.47 | 85.34 | **87.39** |
| InstructBLIP-xxl | 82.53 | 81.87 | 81.23 |
| LlaVa-1.5 | **85.94** | **86.33** | 86.74 |

(d) R-splice

better capture details and nuances in the input data, which enhances their deepfake detection capabilities. Nevertheless, due to the scarcity of high-quality captions and large-scale vision-language datasets tailored to deepfake detection, the improvements remain limited and only in the binary task. Overall, addressing these limitationsby creating specialised datasets and foundation modelscould lead to substantial advancements in this area.

**Limitations and Future Work:** As the models in this work are all evaluated under zero-shot settings, their performance is below that of purpose-build networks seen in previous works [82], particularly for more challenging in-the-wild datasets. This further highlights the need for task specific models and more fine-grained deepfake datasets, which is a key finding of the experiments conducted in this work. A significant limitation is the lack of detailed language descriptions in datasets, making qualitative evaluation harder. Additionally, current datasets lack fine-grained labels, restricting assessments of manipulations to pseudo-fakes and SeqDeepFake [63]. Furthermore, as both the pertaining and evaluation datasets are not unbiased, the performance of all VLLMs is susceptible to the bias of the datasets, which is not addressed in this or previous benchmarks [82]. Identifying these shortcomings is important for future works on the task, particularly as VLLMs gain traction.

# 6 Conclusion

In conclusion, our proposed benchmark has several contributions; first and foremost, we propose a method to transform deepfake detection into a VQA problem beyond binary classification to leverage common sense reasoning as an inherent explainability mechanism. We show how this can be achieved in both a multiple-choice and open-ended VQA –with the latter being the most important use-case for new and unknown face forgery methods. This approach is used to evaluate a multi-label problem that is not typical of classic VQA. By doing so, we can systematically and consistently evaluate the common sense reasoning capabilities of current and future VLLMs in fine-grained deepfake detection.

Our selection of metrics and matching strategies allows for a fair evaluation of the proposed task. In particular, we include metrics that are robust to imbalance in both the binary and multi-label fine-grained tasks. Even though VLLMs in a zero-shot evaluation do not outperform purpose-built methods, the generated responses include reasoning, therefore holding promise for significant contributions in explainable deepfake detection, confirming the initial motivation behind examining the use of such models for the task and understanding the current capabilities. Moreover, as this benchmark can be extended in terms of models and datasets, it allows for a systematic and fair comparison of new language generation methods for explainable deepfake detection.

**Ethics statement:** The authors of this paper acknowledge the crucial role of ethical considerations in AI research and development. Our dedication lies in upholding principles of fairness and impartiality. Recognising the societal implications of generative technology (including VLLMs), we commit to transparency by openly communicating our findings and advancements with the research community.

## Acknowledgments and Disclosure of Funding

This work is supported by the Luxembourg National Research Fund, under the BRIDGES2021/IS/16353350/FaKeDeTeR, and by POST Luxembourg. Experiments were performed on the Luxembourg national supercomputer MeluXina. The authors gratefully acknowledge the LuxProvide teams for their expert support.

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

# Appendix

## A Model Zoo

**LlaVa-1.5** [43] is an extension of the LlaVa [44] model. We use the variant with CLIP-ViT-L-14 as a vision encoder and Vicuna-7b [12] language model. **BLIP-2** [35] uses a QFormer architecture to bridge frozen language and vision encoders. We use the variant with CLIP-ViT-L-14 as a vision encoder and the 2.7b OPT [85] language model. **InstructBLIP** [15] is a family of VLLMs that exploits the basic BLIP-2 [35] architecture and advances the task by giving the instruction to both the QFormer and the LLM. We use two variants of the architecture, with two different LLMs from the T5 family [13]; in the base architecture, we use the CLIP-ViT-L-14 as a vision encoder and the T5-xl LLM. **InstructBLIPxxl** uses the same vision encoder and the T5-xxl language model. A comparison of all architectures in terms of parameters and pre-training datasets can be seen in Tab. 4. It is worth noting, that none of the pre-training datasets are related to deepfakes, making the task more challenging. For the **ensemble**, we chose BLIP-2 [35] and LlaVa-15 [43] , based of two main factors: a) they show the most competitive performance on most datasets as seen in Fig. 3 and b) they have the least overlap in terms of pre-training data thus we intuitively expect them having complementary information. The ensembling method adopted in this work is using score fusion with majority voting; in the occasions where the models disagree, the mean is taken. **GPT4v** is used for comparison in the binary tasks, however as the training details of this model are unknown, it should only be treated as an upper bound.

| Architecture | FLOPS | # Params | Pre-training Data |
|---|---|---|---|
| BLIP-2 [35] | $0.38T$ | $3.74B$ | COCO [42], Visual Genome [33], CC3M [64], CC12M [9], SBU [56], and $115M$ images from the LAION400M [61] |
| InstructBLIP [15] | $0.33T$ | $4.02B$ | COCO [42], WebCapFilt [35], TextCaps [67], VQAv2 [20], OK-VQA [48], AOK-VQA [62], OCR-VQA [51], LLaVA-Instruct-150K [44] |
| InstructBLIP-xxl [15] | $0.53T$ | $12.31B$ | COCO [42], WebCapFilt [35], TextCaps [67], VQAv2 [20], OK-VQA [48], AOK-VQA [62], OCR-VQA [51], LLaVA-Instruct-150K [44] |
| LlaVa-1.5 [43] | $4.14T$ | $7.06B$ | LLaVA-Instruct-150K [44], VQAv2 [20], OK-VQA [48], OCR-VQA [51] |

Table 4: Comparisons of model FLOPS, number of parameters and pre-training datasets for selected VLLMs

**Implementation Details.** All experiments were conducted using four NVIDIA A100 GPUs, with 40GB of memory. We use the PyTorch deep learning framework for all model evaluation tasks and weights published on HuggingFace[5].

## B Binary Classification Prompts

The term deepfake is a colloquial term for a wide range of manipulations using generative models, from altering one small area all the way to fully generated images and videos. As such, the class name itself has several synonyms that can describe it. To assess the model's robustness to instruction, we first prompt an LLM –specifically ChatGPT3.5 to give us synonyms for a deepfake. This is done as an automation step to incorporate the general consensus into the method without the author's bias, following previous works [50]. The following synonyms are tested for the positive class: "manipulated", "deepfake", "synthetic", "altered", "fabricated", "face forgery" and "falsified". We show the detailed performance of all models on all synonyms in Tab. 6. To provide context, we also provide the cross-dataset performance of several discriminative SOTA works in Tab. 5. The models show the most consistent performance on synonyms "manipulated", "synthetic" and "altered"; therefore, we do all subsequent analyses on these prompts.

---

[5]https://huggingface.co/

Table 5: Reported cross-dataset performance of purpose-built discriminative SoTA models.

| | CelebDF [41] | | DFW [90] | | DFDC [17] | |
|---|---|---|---|---|---|---|
| | AUC | mAP | AUC | mAP | AUC | mAP |
| UCF [81] | 82.40 | - | - | - | 80.50 | - |
| X-Ray [37] | 79.50 | - | - | - | 65.50 | - |
| Xception [60] | 61.18 | 66.93 | 65.29 | 55.37 | 69.90 | 91.98 |
| REECE [7] | 70.93 | 70.35 | 68.16 | 54.41 | - | - |
| CORE [53] | 74.28 | - | - | - | 73.41 | - |
| LAA-Net [52] | 86.28 | 91.93 | 57.13 | 56.89 | 69.69 | 93.67 |

## C Vision Encoder Fine-tuning

We fine-tune the CLIP-L/14-336 Vision encoder using LoRA [23] adaptors. Specifically, we add 32 adaptors to the queries, keys, values and out projection of the attention heads, with an alpha of 32, a dropout rate of 0.2 and 4bit quantization. As the available datasets lack sample level descriptions, we use the text embeddings of the synonyms for the positive category and the embeddings of "real", "original", "unaltered", "authentic", "legitimate", "genuine", "bona fide" for the negative. We apply a Sigmoid loss over the cosine similarity [83] of all relevant text embeddings in the batch and train the vision encoder for 50 epochs. The updated weights are then used in the LlaVa-1.5 architecture.

## D CLIP Embeddings

As most models use CLIP [59] variants as backbone vision encoders, we assess the separability of samples in real and manipulated images in each dataset by visualising them in a two-dimensional plane using t-SNE. The resulting visualisations can be seen in Fig. 5. The samples appear more separable in some datasets. Interestingly, the StyleGAN datasets, where the images are fully generated, seem to have more distinguishable latent representations from real images. The separability is not necessarily reflected in the language generation as seen in Sec. 5.1; to further examine the root cause of this, we first calculate the average image embedding of each class so as to create a class prototype and retrieve the top-10 nearest language token embeddings, seen in Tab. 8. The first observation that can be made is that none of the tokens seems related to the task at hand, which would potentially inform prompt selection, so without the reasoning capabilities of the LLM, the token retrieval on its own is not very informative. Secondly, we see that a number of tokens are repeated across datasets and for both classes, which can be attributed to the much smaller sub-space of the deepfake task compared to the CLIP latent space; so, while the image embeddings are somewhat separable for several datasets retrieving the nearest language tokens shows that the subspace is not very much related to face forgery. From these two observations, we can better understand the zero-shot performance of the tested foundation models. Finally, we show the performance of CLIP on the binary classification task in Tab. 7. We use an ensemble of prompts using the Imagenet prompt templates [59]; for the positive class, we average the embeddings of the prompts for synonyms "manipulated", "synthetic" and "altered"; for the negative, we use "real", "original" and "unaltered". Contrary to previous VQA works [14, 19] that use CLIP retrieval as an upper bound, we see this is not the case in the task. This can be attributed to the more abstract definition of the class in the deepfake detection task, as well as the pre-training dataset, which is more object-oriented.

## E Human Evaluation

The human evaluation is based upon previous studies in VQA evaluation [19, 10]. The humans are asked to rate model predictions on a scale from 1 to 5, where 1 is completely wrong and 5 is completely correct.

**Evaluation dataset:** We use a subset of the pseudo-fake R-Splicer dataset for the human evaluation study, as it has artefacts visible to the human eye and is thus easier for the annotators to assess the response quality.

Table 6: Binary Performance of tested models

**(a) BLIP-2 [35] performance on the nine benchmark datasets.**

| Synonym | Seq. Deepfake Attr. | | | Seq. Deepfake Comp. | | | R-Splicer | | | FF++ | | | DFDC | | | CelebDF | | | DFW | | | StyleGAN2 | | | StyleGAN3 | | |
|---|---|---|---|---|---|---|---|---|---|---|---|---|---|---|---|---|---|---|---|---|---|---|---|---|---|---|---|
| | Acc | AUC | F1 | Acc | AUC | F1 | Acc | AUC | F1 | Acc | AUC | F1 | Acc | AUC | F1 | Acc | AUC | F1 | Acc | AUC | F1 | Acc | AUC | F1 | Acc | AUC | F1 |
| manipulated | 95.36 | 50.49 | 97.62 | 94.44 | 50.46 | 97.14 | 95.32 | 50.59 | 97.60 | 65.42 | 49.76 | 79.00 | 49.47 | 49.95 | 66.13 | 52.20 | 49.87 | 68.34 | 50.40 | 49.99 | 66.93 | 52.60 | 50.25 | 68.69 | 50.00 | 49.29 | 66.33 |
| altered | 96.06 | 50.25 | 97.99 | 94.62 | 49.93 | 97.23 | 95.85 | 50.22 | 97.88 | 65.76 | 50.01 | 79.24 | 49.50 | 49.98 | 66.21 | 51.60 | 49.31 | 67.73 | 50.37 | 49.96 | 66.94 | 53.20 | 50.84 | 69.13 | 51.49 | 50.76 | 67.66 |
| synthetic | 95.48 | 51.75 | 97.68 | 93.95 | 52.09 | 96.87 | 90.45 | 54.42 | 94.95 | 65.27 | 51.63 | 78.15 | 49.46 | 49.93 | 65.95 | 40.80 | 39.20 | 56.21 | 50.51 | 50.17 | 65.05 | 54.60 | 52.37 | 69.53 | 53.73 | 53.05 | 68.37 |
| deepfake | 93.92 | 51.54 | 96.86 | 93.15 | 54.50 | 96.43 | 89.47 | 54.91 | 94.40 | 64.66 | 52.19 | 77.28 | 49.35 | 49.82 | 65.81 | 39.40 | 37.79 | 55.24 | 50.48 | 50.16 | 64.18 | 53.80 | 51.53 | 69.16 | 50.75 | 50.07 | 66.33 |
| fabricated | 95.00 | 51.80 | 97.43 | 94.02 | 50.56 | 96.92 | 91.70 | 52.53 | 95.65 | 64.79 | 50.58 | 78.04 | 49.41 | 49.89 | 66.03 | 46.60 | 44.64 | 62.66 | 50.85 | 50.49 | 65.80 | 53.80 | 51.51 | 69.24 | 51.49 | 50.80 | 67.01 |
| face forgery | 96.11 | 49.98 | 98.01 | 94.72 | 51.87 | 97.28 | 94.18 | 51.18 | 97.00 | 65.03 | 50.42 | 78.36 | 49.44 | 49.92 | 66.01 | 53.00 | 50.69 | 68.79 | 49.67 | 49.27 | 66.19 | 53.60 | 51.26 | 69.31 | 51.49 | 50.76 | 67.66 |
| falsified | 95.58 | 52.10 | 97.73 | 93.29 | 51.74 | 96.52 | 92.03 | 53.25 | 95.83 | 65.42 | 51.46 | 78.38 | 49.40 | 49.88 | 65.94 | 42.80 | 41.07 | 58.55 | 52.92 | 52.56 | 67.06 | 53.80 | 51.57 | 68.99 | 52.99 | 52.30 | 68.02 |

**(b) InstructBLIP [15] performance on the nine benchmark datasets.**

| Synonym | Seq. Deepfake Attr. | | | Seq. Deepfake Comp. | | | R-Splicer | | | FF++ | | | DFDC | | | CelebDF | | | DFW | | | StyleGAN2 | | | StyleGAN3 | | |
|---|---|---|---|---|---|---|---|---|---|---|---|---|---|---|---|---|---|---|---|---|---|---|---|---|---|---|---|
| | Acc | AUC | F1 | Acc | AUC | F1 | Acc | AUC | F1 | Acc | AUC | F1 | Acc | AUC | F1 | Acc | AUC | F1 | Acc | AUC | F1 | Acc | AUC | F1 | Acc | AUC | F1 |
| manipulated | 43.85 | 57.87 | 59.50 | 84.34 | 69.39 | 91.25 | 80.95 | 76.04 | 89.17 | 53.87 | 60.01 | 54.01 | 55.11 | 54.90 | 41.78 | 19.40 | 19.95 | 9.84 | 58.73 | 58.68 | 61.10 | 17.40 | 18.05 | 5.49 | 26.12 | 26.25 | 19.51 |
| altered | 45.10 | 55.92 | 60.86 | 82.10 | 65.58 | 89.91 | 81.31 | 76.41 | 89.40 | 54.32 | 61.01 | 53.83 | 55.51 | 55.29 | 42.39 | 19.40 | 19.94 | 10.24 | 58.37 | 58.32 | 61.23 | 17.80 | 18.41 | 6.80 | 27.61 | 27.70 | 23.62 |
| synthetic | 74.33 | 61.96 | 84.97 | 88.39 | 70.21 | 93.67 | 88.14 | 62.50 | 93.61 | 59.24 | 54.51 | 69.17 | 53.49 | 53.79 | 64.10 | 44.20 | 42.93 | 56.61 | 51.54 | 51.23 | 64.65 | 29.60 | 28.99 | 38.25 | 38.06 | 37.86 | 45.75 |
| deepfake | 20.19 | 58.62 | 29.42 | 62.94 | 75.23 | 75.93 | 66.76 | 78.77 | 79.26 | 49.04 | 60.62 | 39.09 | 53.54 | 53.17 | 24.19 | 26.80 | 28.13 | 0.54 | 59.07 | 59.30 | 43.77 | 26.80 | 28.13 | 0.54 | 28.36 | 28.74 | 4.00 |
| fabricated | 74.23 | 58.01 | 84.96 | 96.08 | 63.74 | 97.97 | 91.92 | 57.64 | 95.76 | 62.85 | 50.97 | 75.76 | 52.18 | 52.42 | 61.59 | 55.80 | 53.82 | 69.26 | 51.35 | 51.03 | 65.02 | 28.60 | 27.87 | 38.77 | 32.84 | 32.53 | 44.44 |
| face forgery | 19.90 | 58.47 | 28.99 | 54.55 | 74.75 | 68.60 | 71.10 | 82.75 | 82.42 | 51.26 | 62.73 | 42.42 | 54.76 | 54.42 | 30.14 | 26.80 | 28.13 | 0.54 | 56.81 | 57.03 | 41.85 | 27.00 | 28.32 | 1.08 | 32.84 | 33.29 | 4.26 |
| falsified | 19.62 | 58.33 | 28.55 | 57.76 | 75.13 | 71.51 | 64.94 | 81.28 | 77.80 | 49.44 | 61.70 | 38.24 | 54.15 | 53.80 | 27.19 | 26.00 | 27.29 | 0.54 | 55.86 | 56.14 | 34.03 | 26.00 | 27.29 | 0.54 | 29.10 | 29.52 | 2.06 |

**(c) InstructBLIP [15] with Flan-T5-xxl [13] language model performance on the nine benchmark datasets.**

| Synonym | Seq. Deepfake Attr. | | | Seq. Deepfake Comp. | | | R-Splicer | | | FF++ | | | DFDC | | | CelebDF | | | DFW | | | StyleGAN2 | | | StyleGAN3 | | |
|---|---|---|---|---|---|---|---|---|---|---|---|---|---|---|---|---|---|---|---|---|---|---|---|---|---|---|---|
| | Acc | AUC | F1 | Acc | AUC | F1 | Acc | AUC | F1 | Acc | AUC | F1 | Acc | AUC | F1 | Acc | AUC | F1 | Acc | AUC | F1 | Acc | AUC | F1 | Acc | AUC | F1 |
| manipulated | 16.44 | 56.68 | 23.57 | 58.88 | 77.04 | 72.42 | 58.49 | 76.85 | 72.63 | 48.53 | 60.88 | 36.70 | 53.88 | 53.54 | 28.75 | 24.80 | 26.05 | 0.00 | 54.09 | 54.37 | 32.36 | 24.80 | 26.05 | 0.00 | 25.37 | 25.74 | 1.96 |
| altered | 17.21 | 57.08 | 24.80 | 60.98 | 78.14 | 74.19 | 60.99 | 77.24 | 74.72 | 49.04 | 61.25 | 37.78 | 53.99 | 53.66 | 29.59 | 23.40 | 24.52 | 1.54 | 54.90 | 55.15 | 36.37 | 23.20 | 24.33 | 1.03 | 38.81 | 39.39 | 0.00 |
| synthetic | 17.40 | 57.18 | 25.11 | 56.64 | 77.17 | 70.42 | 59.76 | 75.42 | 73.74 | 49.74 | 61.65 | 39.46 | 53.79 | 53.44 | 27.57 | 24.00 | 25.21 | 0.00 | 58.53 | 58.67 | 57.14 | 24.00 | 25.20 | 0.52 | 28.36 | 28.74 | 4.00 |
| deepfake | 7.98 | 52.29 | 8.77 | 13.43 | 54.42 | 16.24 | 41.08 | 69.28 | 56.04 | 42.22 | 56.31 | 22.40 | 52.28 | 51.85 | 14.48 | 35.00 | 36.76 | 0.00 | 51.36 | 51.73 | 13.75 | 35.00 | 36.76 | 0.00 | 38.81 | 39.39 | 0.00 |
| fabricated | 21.06 | 59.07 | 30.72 | 64.34 | 79.91 | 76.92 | 64.97 | 77.75 | 77.90 | 51.71 | 62.81 | 43.75 | 54.40 | 54.08 | 31.55 | 23.80 | 24.96 | 1.04 | 56.00 | 56.13 | 48.37 | 23.60 | 24.77 | 0.52 | 25.37 | 25.74 | 1.96 |
| face forgery | 9.42 | 53.04 | 11.47 | 30.63 | 63.48 | 42.46 | 41.83 | 69.76 | 56.84 | 42.83 | 56.74 | 23.58 | 52.24 | 51.82 | 16.20 | 34.20 | 35.92 | 0.00 | 51.66 | 52.02 | 15.64 | 34.20 | 35.92 | 0.00 | 26.87 | 27.25 | 2.00 |
| falsified | 16.73 | 56.83 | 24.04 | 58.74 | 78.28 | 72.25 | 63.55 | 78.75 | 76.76 | 49.68 | 61.77 | 38.94 | 53.81 | 53.47 | 28.49 | 24.40 | 25.57 | 1.56 | 54.25 | 54.49 | 35.95 | 24.00 | 25.19 | 0.52 | 26.87 | 27.25 | 2.00 |

**(d) LlaVa-15 [43] performance on the nine benchmark datasets.**

| Synonym | Seq. Deepfake Attr. | | | Seq. Deepfake Comp. | | | R-Splicer | | | FF++ | | | DFDC | | | CelebDF | | | DFW | | | StyleGAN2 | | | StyleGAN3 | | |
|---|---|---|---|---|---|---|---|---|---|---|---|---|---|---|---|---|---|---|---|---|---|---|---|---|---|---|---|
| | Acc | AUC | F1 | Acc | AUC | F1 | Acc | AUC | F1 | Acc | AUC | F1 | Acc | AUC | F1 | Acc | AUC | F1 | Acc | AUC | F1 | Acc | AUC | F1 | Acc | AUC | F1 |
| manipulated | 73.05 | 57.79 | 84.14 | 94.34 | 53.56 | 97.07 | 94.05 | 62.20 | 96.90 | 67.52 | 55.20 | 79.18 | 54.51 | 54.80 | 64.63 | 38.80 | 37.47 | 52.78 | 52.92 | 52.58 | 66.54 | 37.40 | 36.13 | 51.17 | 47.01 | 46.46 | 61.62 |
| altered | 89.16 | 55.66 | 94.22 | 94.30 | 51.65 | 97.06 | 96.10 | 52.45 | 98.01 | 66.42 | 51.41 | 79.42 | 50.67 | 51.11 | 65.95 | 49.00 | 46.85 | 65.31 | 51.05 | 50.65 | 67.17 | 47.20 | 45.12 | 63.74 | 50.75 | 50.04 | 66.67 |
| synthetic | 14.54 | 54.96 | 20.09 | 65.21 | 79.44 | 77.59 | 71.17 | 75.60 | 82.58 | 59.69 | 65.70 | 60.69 | 56.87 | 56.10 | 60.69 | 44.20 | 39.98 | 7.34 | 56.08 | 56.11 | 50.97 | 16.40 | 17.19 | 0.95 | 21.64 | 21.86 | 8.70 |
| deepfake | 8.00 | 52.16 | 8.29 | 38.46 | 67.53 | 51.91 | 51.03 | 74.43 | 65.99 | 46.16 | 59.25 | 31.44 | 54.18 | 53.79 | 23.32 | 34.60 | 36.33 | 0.61 | 53.63 | 53.98 | 20.52 | 34.20 | 35.92 | 0.00 | 37.31 | 37.88 | 0.00 |
| fabricated | 12.72 | 54.31 | 16.93 | 58.11 | 75.38 | 71.73 | 66.03 | 78.94 | 78.70 | 51.93 | 63.19 | 43.73 | 54.71 | 54.26 | 0.53 | 53.82 | 54.07 | 33.94 | 25.00 | 26.24 | 0.53 | 39.81 | 48.39 | 64.50 | 55.97 | 65.32 | 9.62 |
| face forgery | 12.62 | 54.56 | 16.72 | 56.78 | 76.25 | 70.50 | 71.86 | 84.60 | 82.94 | 55.75 | 66.12 | 50.36 | 56.38 | 56.05 | 33.71 | 29.40 | 30.79 | 2.75 | 55.02 | 55.29 | 35.12 | 28.60 | 30.02 | 0.56 | 32.84 | 33.31 | 2.17 |
| falsified | 16.85 | 56.46 | 23.86 | 64.83 | 77.98 | 77.32 | 78.19 | 84.70 | 87.30 | 60.02 | 68.31 | 58.50 | 56.98 | 56.72 | 40.85 | 23.00 | 23.93 | 5.87 | 57.44 | 57.62 | 46.59 | 21.40 | 22.42 | 1.50 | 26.87 | 27.18 | 7.55 |

**(e) Ensemble performance on the nine benchmark datasets.**

| Synonym | Seq. Deepfake Attr. | | | Seq. Deepfake Comp. | | | R-Splicer | | | FF++ | | | DFDC | | | CelebDF | | | DFW | | | StyleGAN2 | | | StyleGAN3 | | |
|---|---|---|---|---|---|---|---|---|---|---|---|---|---|---|---|---|---|---|---|---|---|---|---|---|---|---|---|
| | Acc | AUC | F1 | Acc | AUC | F1 | Acc | AUC | F1 | Acc | AUC | F1 | Acc | AUC | F1 | Acc | AUC | F1 | Acc | AUC | F1 | Acc | AUC | F1 | Acc | AUC | F1 |
| manipulated | 95.38 | 58.70 | 97.63 | 94.83 | 67.61 | 97.34 | 96.24 | 54.77 | 98.08 | 66.33 | 74.05 | 79.69 | 49.61 | 54.59 | 66.18 | 52.40 | 51.21 | 68.60 | 50.50 | 59.84 | 67.05 | 53.79 | 58.98 | 68.39 | 50.75 | 50.00 | 67.33 |
| altered | 95.73 | 60.44 | 97.82 | 94.83 | 47.48 | 97.34 | 96.40 | 52.40 | 98.17 | 66.18 | 83.08 | 79.63 | 49.57 | 60.67 | 66.24 | 52.40 | 50.00 | 68.77 | 50.42 | 50.00 | 67.04 | 55.36 | 72.93 | 69.86 | 50.75 | 50.00 | 67.33 |
| synthetic | 80.63 | 54.15 | 89.02 | 65.31 | 56.09 | 77.70 | 88.48 | 53.20 | 93.83 | 63.24 | 56.20 | 74.45 | 51.09 | 52.60 | 63.29 | 44.20 | 39.57 | 7.14 | 48.37 | 46.35 | 61.71 | 58.20 | 58.93 | 66.07 | 55.22 | 64.14 | 68.42 |
| deepfake | 64.90 | 51.64 | 78.13 | 37.76 | 53.74 | 51.26 | 88.17 | 52.14 | 93.66 | 63.12 | 54.90 | 75.02 | 49.72 | 50.65 | 64.95 | 43.40 | 37.76 | 57.06 | 49.84 | 49.01 | 63.39 | 53.15 | 52.78 | 61.87 | 50.75 | 50.42 | 64.52 |
| fabricated | 94.62 | 52.63 | 97.23 | 13.17 | 51.97 | 18.13 | 91.13 | 53.02 | 95.33 | 64.09 | 56.23 | 75.85 | 49.80 | 56.09 | 66.35 | 51.80 | 50.20 | 64.50 | | | | 55.05 | 55.36 | 64.60 | 55.97 | 65.32 | 9.62 |
| face forgery | 75.96 | 49.16 | 86.26 | 12.60 | 51.96 | 17.14 | 94.23 | 54.09 | 97.02 | 65.67 | 58.53 | 77.33 | 51.12 | 56.03 | 65.74 | 55.20 | 69.48 | 69.81 | 48.77 | 38.75 | 65.08 | 44.79 | 42.94 | 54.55 | 48.51 | 44.79 | 63.10 |
| falsified | 94.42 | 52.28 | 97.13 | 63.64 | 55.34 | 76.45 | 93.91 | 55.39 | 96.85 | 66.58 | 60.22 | 78.59 | 49.72 | 51.56 | 65.69 | 44.40 | 36.23 | 59.36 | 52.09 | 55.80 | 66.32 | 50.63 | 49.29 | 61.88 | 52.99 | 59.33 | 67.36 |

**Human evaluation:** We select 100 samples from the dataset and generate responses using the open-ended prompt. Each sample is annotated by the three human annotators on a scale from 1 to 5, where 1 is completely wrong, and 5 is completely correct. Each annotator is shown 50 samples. To reduce the workload on human annotators, we assign 1 (completely wrong) to all responses that describe the image content but no areas of manipulation. The annotations are then standardised between 0 and 1. An example of the form shown to human evaluators can be seen in Fig. 6.

**Annotator agreement:** as in previous works [19, 10] we use Krippendorff's alpha to assess the inter-annotator agreement and obtain .75, which is a strong agreement given the complexity of the multi-label task. We average the annotator scores to receive the final gold standard for each sample and then obtain the average human score for each model.

# F  Qualitative Samples

By examining a few samples in Fig. 7, we can also see that all models tend to hallucinate or provide a general description of the image content when they fail to identify specific areas of manipulation. Furthermore, while BLIP-2 [35] tends to respond more concisely –which is identified by Liu *et al.*

Table 7: CLIP baseline performance on the binary task of the nine selected benchmarks.

| Dataset | Accuracy | AUC | F1 |
|---|---|---|---|
| SeqDeepFake attributes [63] | 54.62 | 63.46 | 69.63 |
| SeqDeepFake components [63] | 35.80 | 49.10 | 50.38 |
| R-splice | 17.14 | 44.33 | 25.98 |
| FF++ [60] | 34.85 | 46.50 | 17.38 |
| DFDC [17] | 46.84 | 46.75 | 40.93 |
| CelebDF [41] | 59.80 | 60.95 | 49.11 |
| DFW [90] | 49.67 | 49.90 | 31.09 |
| StyleGan2 [28]. | 69.20 | 69.92 | 65.16 |
| StyleGan3 [29] | 69.40 | 69.63 | 64.35 |

Table 8: Top 10 closest tokens to the class prototypes. Unique to the class tokens are in **bold**.

| Dataset | Original | DeepFake |
|---|---|---|
| SeqDeepFake attributes | natives, labeling, liz, **saving**, rink, **demon**, pitch, creole, godis, wentz | natives, liz, labeling, rink, wentz, **ronda**, godis, creole, **%**), melissa |
| SeqDeepFake components | natives, **labeling**, qld, romo, liz, **creole**, **dhoni**, **.**, **hijab**, gaining | natives, qld, liz, gaining, romo, **%**), **cuomo**, **klo**, **minions**, **cowgirl** |
| R-splice | natives, wentz, anglo, liz, %), qld, anca, **romo**, **ronda**, ural | anglo, %), wentz, liz, natives, **klo**, anca, qld, ural, **weed** |
| FF++ | natives, qld, wentz, anglo, cuomo, anca, liz, **labeling**, romo, **melissa** | natives, qld, wentz, anglo, liz, cuomo, anca, **%**), **weed**, romo |
| DFDC | natives, wentz, anglo, liz, ronda, qld, %), anca, romo, )!! | anglo, wentz, natives, liz, %), ronda, qld, anca, )!!, romo |
| CelebDF | natives, **liz**, wentz, **ronda**, %), anglo, **qld**, romo, **labeling**, **anca** | anglo, natives, **ural**, **klo**, **creole**, %), **weed**, wentz, romo, **minions** |
| DFW | anglo, %), ronda, wentz, ural, natives, **liz**, klo, **rene**, ator | anglo, %), natives, ural, wentz, ator, **ronda**, klo, **anca**, qld |
| StyleGan2 | natives, liz, wentz, ronda, **%**), **anglo**, **qld**, romo, labeling, **anca** | natives, labeling, liz, **rink**, ronda, romo, **creole**, **melissa**, wentz, **saving** |
| StyleGan3 | natives, liz, wentz, ronda, romo, labeling, **aa**, **anca**, **%**), **ural** | natives, labeling, liz, ronda, **rink**, wentz, romo, **saving**, **creole**, **melissa** |

[44] as well–, we note that larger models attempt to provide a justification for their response, which is ultimately the goal of investigating VLLMs for this task.

As the available datasets lack sample specific descriptions of manipulation areas, we include BertScore of the tested VLLMs on the MagicBrush [84] dataset to assess response quality on the neighbouring task of image editing detection, that is language driven in table Tab. 9.

# G   Multiple-choice VQA

Further to the analysis in Sec. 5.2, we show the performance of all models on the multiple-choice VQA setting in Tab. 10. In the multiple-choice setting, all models have comparable mAP and AUC, however, LlaVa-1.5 [43] shows a clear advantage in terms of Recall and F1.

# H   Open-ended VQA

The detailed performance of each model on the open-ended fine-graned detection can be seen on Tabs. 11 to 14. The matching strategy has a significant impact on Recall, as mentioned in Sec. 5.2, thus also increasing the F1 score.

Table 9: BertScore of selected VLLMs on MagicBrush dataset

| | Precision | Recall | F1 |
|---|---|---|---|
| Blip | 85.44 | 84.37 | 84.57 |
| InstructBlip | 84.56 | 85.09 | 84.82 |
| InstructBlip-xxl | 81.07 | 84.55 | 81.88 |
| LlaVa 1.5 | 83.08 | 84.77 | 83.87 |

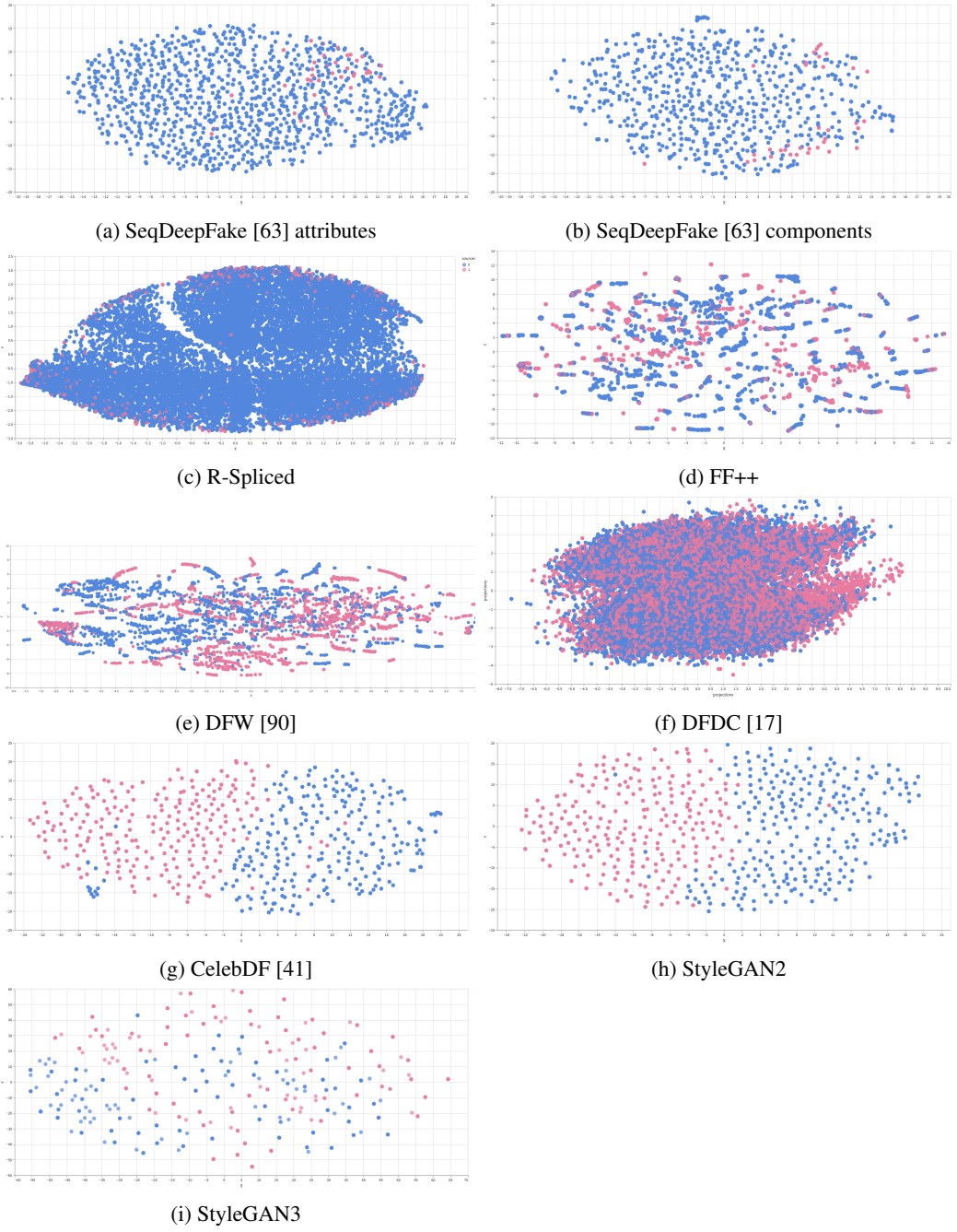

Figure 5: t-SNE Visualisation of CLIP [59] image embeddings on the test set of the selected datasets (perplexity=50)

## Qualitative evaluation

The aim of the study is to understand how well the generated responses reflect the ground truth i.e. how well each model can spot the manipulated areas.

The possible manipulation areas are: entire face (faceswap), mouth, nose, eyes, eyebrows Each model is instructed to "Explain briefly what areas of this image are manipulated?".

See the image, the correct answer and the predicted answers.
Select the score that best reflects how closely the predicted answers capture the same information as the correct answer
This means that the statements should be semantically similar but do not need to match the class names exactly.

1. completely wrong
2. mostly wrong
3. half right
4. mostly right
5. completely right

The entire evaluation should take 30-60 mins.

The areas that are manipulated are: eyebrows mouth faceswap

The eyes of the man in the hat

|  | 1 | 2 | 3 | 4 | 5 |  |
|---|---|---|---|---|---|---|
| completely wrong | ○ | ○ | ○ | ○ | ○ | completely correct |

(a) Annotator Briefing  (b) Annotation Form

Figure 6: Example of Briefing( a) and Annotation Form( b) shown to human evaluators.

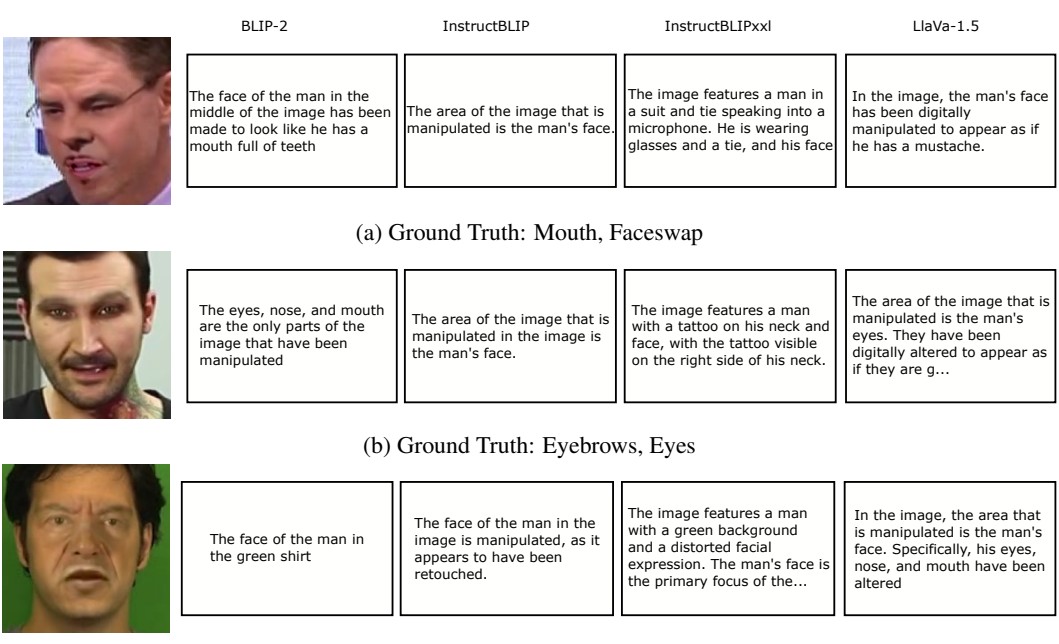

| | BLIP-2 | InstructBLIP | InstructBLIPxxl | LlaVa-1.5 |
|---|---|---|---|---|
| | The face of the man in the middle of the image has been made to look like he has a mouth full of teeth | The area of the image that is manipulated is the man's face. | The image features a man in a suit and tie speaking into a microphone. He is wearing glasses and a tie, and his face | In the image, the man's face has been digitally manipulated to appear as if he has a mustache. |

(a) Ground Truth: Mouth, Faceswap

| | The eyes, nose, and mouth are the only parts of the image that have been manipulated | The area of the image that is manipulated in the image is the man's face. | The image features a man with a tattoo on his neck and face, with the tattoo visible on the right side of his neck. | The area of the image that is manipulated is the man's eyes. They have been digitally altered to appear as if they are g... |

(b) Ground Truth: Eyebrows, Eyes

| | The face of the man in the green shirt | The face of the man in the image is manipulated, as it appears to have been retouched. | The image features a man with a green background and a distorted facial expression. The man's face is the primary focus of the... | In the image, the area that is manipulated is the man's face. Specifically, his eyes, nose, and mouth have been altered |

(c) Ground Truth: Nose, Mouth, Faceswap

Figure 7: Samples of generated responses in an open-ended setting.

Table 10: Performance of tested VLLMs in multiple-choice setting

|  | AUC | F1 | mAP | Recall |  | AUC | F1 | mAP | Recall |  | AUC | F1 | mAP | Recall |
|---|---|---|---|---|---|---|---|---|---|---|---|---|---|---|
| Bangs | 49.8 | 0.0 | 60.3 | 0.0 | nose | 50.0 | 0.0 | 55.4 | 0.0 | nose | 51.3 | 46.9 | 52.5 | 42.2 |
| Eyeglasses | 49.9 | 8.0 | 63.4 | 4.6 | eye | 50.8 | 10.2 | 66.8 | 5.5 | eyebrows | 50.4 | 57.0 | 52.0 | 63.1 |
| Beard | 49.8 | 1.0 | 61.0 | 0.5 | eyebrow | 51.0 | 11.0 | 58.3 | 6.0 | eyes | 50.2 | 56.2 | 51.5 | 61.9 |
| Smiling | 50.0 | 0.0 | 58.3 | 0.0 | lip | 50.0 | 0.0 | 67.6 | 0.0 | mouth | 49.9 | 45.4 | 51.9 | 40.9 |
| Young | 50.0 | 0.0 | 62.3 | 0.0 | hair | 50.4 | 9.7 | 48.9 | 5.4 | faceswap | 44.6 | 9.2 | 50.2 | 5.5 |
| Total | 49.9 | 4.5 | 61.1 | 2.5 | Total | 50.4 | 10.3 | 59.4 | 5.7 | Total | 49.3 | 42.9 | 51.6 | 42.7 |

(a) SeqDeepFake attributes [63]    (b) SeqDeepFake comp [63]    (c) R-Splicer

BLIP-2 [35]

|  | AUC | F1 | mAP | Recall |  | AUC | F1 | mAP | Recall |  | AUC | F1 | mAP | Recall |
|---|---|---|---|---|---|---|---|---|---|---|---|---|---|---|
| Bangs | 50.2 | 1.0 | 60.4 | 0.5 | nose | 50.0 | 0.0 | 55.4 | 0.0 | nose | 56.5 | 38.5 | 56.2 | 30.4 |
| Eyeglasses | 58.4 | 51.3 | 68.0 | 41.6 | eye | 52.2 | 11.7 | 67.7 | 6.5 | eyebrows | 52.5 | 58.0 | 53.1 | 66.5 |
| Beard | 50.0 | 0.3 | 61.0 | 0.2 | eyebrow | 50.6 | 3.3 | 58.2 | 1.7 | eyes | 52.6 | 16.5 | 53.1 | 10.1 |
| Smiling | 50.0 | 0.0 | 58.3 | 0.0 | lip | 50.0 | 0.0 | 67.6 | 0.0 | mouth | 52.9 | 13.0 | 54.3 | 7.1 |
| Young | 50.4 | 29.0 | 62.6 | 21.4 | hair | 51.1 | 24.7 | 49.3 | 23.6 | faceswap | 50.0 | 0.0 | 51.6 | 0.0 |
| Total | 51.8 | 20.4 | 62.1 | 15.9 | Total | 50.8 | 13.2 | 59.6 | 10.6 | Total | 52.9 | 31.5 | 53.7 | 28.5 |

(d) SeqDeepFake attributes [63]    (e) SeqDeepFake comp [63]    (f) R-Splicer

InstructBLIP [15]

|  | AUC | F1 | mAP | Recall |  | AUC | F1 | mAP | Recall |  | AUC | F1 | mAP | Recall |
|---|---|---|---|---|---|---|---|---|---|---|---|---|---|---|
| Bangs | 50.4 | 18.5 | 60.7 | 13.2 | nose | 50.0 | 0.0 | 55.4 | 0.0 | nose | 53.4 | 59.0 | 53.6 | 64.6 |
| Eyeglasses | 54.3 | 75.9 | 65.7 | 90.3 | eye | 57.3 | 39.1 | 70.3 | 31.2 | eyebrows | 52.7 | 31.5 | 53.3 | 22.9 |
| Beard | 50.0 | 0.1 | 61.0 | 0.1 | eyebrow | 54.4 | 21.0 | 60.8 | 13.0 | eyes | 51.0 | 8.5 | 52.0 | 4.7 |
| Smiling | 50.0 | 0.0 | 58.3 | 0.0 | lip | 50.0 | 0.0 | 67.6 | 0.0 | mouth | 57.0 | 41.6 | 56.5 | 32.3 |
| Young | 50.0 | 0.0 | 62.3 | 0.0 | hair | 50.1 | 38.4 | 48.8 | 44.4 | faceswap | 50.0 | 1.2 | 51.6 | 0.6 |
| Total | 50.9 | 31.5 | 61.6 | 34.5 | Total | 52.4 | 32.8 | 60.6 | 29.5 | Total | 52.8 | 28.4 | 53.4 | 25.0 |

(g) SeqDeepFake attributes [63]    (h) SeqDeepFake comp [63]    (i) R-Splicer

InstructBLIP [15] with T5-xxl [13] LLM

|  | AUC | F1 | mAP | Recall |  | AUC | F1 | mAP | Recall |  | AUC | F1 | mAP | Recall |
|---|---|---|---|---|---|---|---|---|---|---|---|---|---|---|
| Bangs | 52.2 | 38.1 | 61.4 | 27.3 | nose | 50.0 | 3.2 | 55.4 | 1.7 | nose | 49.6 | 67.6 | 51.6 | 98.2 |
| Eyeglasses | 58.2 | 73.0 | 67.5 | 77.8 | eye | 48.8 | 76.2 | 65.9 | 91.0 | eyebrows | 50.1 | 67.9 | 51.8 | 98.4 |
| Beard | 57.1 | 74.3 | 64.7 | 86.7 | eyebrow | 49.2 | 71.8 | 57.4 | 96.0 | eyes | 50.2 | 67.7 | 51.5 | 98.9 |
| Smiling | 54.7 | 36.4 | 61.1 | 25.2 | lip | 49.5 | 3.8 | 67.5 | 2.0 | mouth | 50.0 | 68.0 | 51.9 | 98.4 |
| Young | 49.5 | 74.1 | 62.1 | 92.6 | hair | 52.2 | 65.4 | 49.9 | 94.9 | faceswap | 50.4 | 68.2 | 51.8 | 99.8 |
| Total | 54.3 | 59.2 | 63.3 | 61.9 | Total | 49.9 | 44.1 | 59.2 | 57.1 | Total | 50.1 | 67.9 | 51.7 | 98.7 |

(j) SeqDeepFake attributes [63]    (k) SeqDeepFake comp [63]    (l) R-Splicer

LlaVa-1.5 [43]

Table 11: BLIP-2 [35] performance with *contains* and *CLIP* matching in open-ended VQA

| | AUC | F1 | mAP | Recall |
|---|---|---|---|---|
| Bangs | 53.9 | 33.7 | 62.5 | 22.2 |
| Eyeglasses | 56.4 | 40.3 | 67.0 | 28.0 |
| Beard | 50.1 | 2.1 | 61.1 | 1.1 |
| Smiling | 49.9 | 1.0 | 58.3 | 0.5 |
| Young | 44.6 | 24.9 | 60.3 | 18.2 |
| Total | 51.0 | 20.4 | 61.8 | 14.0 |

(a) SeqDeepFake attributes [63] with *contains* matching

| | AUC | F1 | mAP | Recall |
|---|---|---|---|---|
| Bangs | 56.0 | 73.5 | 64.1 | 99.9 |
| Eyeglasses | 60.0 | 75.9 | 69.6 | 99.9 |
| Beard | 59.9 | 74.1 | 66.8 | 99.9 |
| Smiling | 50.2 | 72.0 | 56.4 | 99.9 |
| Young | 51.6 | 75.1 | 61.2 | 100.0 |
| Total | 55.5 | 74.1 | 63.6 | 99.9 |

(b) SeqDeepFake attributes [63] with *CLIP* matching

| | AUC | F1 | mAP | Recall |
|---|---|---|---|---|
| nose | 49.4 | 20.1 | 55.2 | 13.2 |
| eye | 50.1 | 10.5 | 66.6 | 6.0 |
| eyebrow | 49.9 | 2.2 | 57.7 | 1.1 |
| lip | 50.6 | 11.6 | 68.0 | 6.5 |
| hair | 52.6 | 29.3 | 50.3 | 19.8 |
| Total | 50.5 | 14.7 | 59.5 | 9.3 |

(c) SeqDeepFake comp. [63] with *contains* matching

| | AUC | F1 | mAP | Recall |
|---|---|---|---|---|
| nose | 51.6 | 69.0 | 54.8 | 100.0 |
| eye | 49.8 | 77.3 | 64.1 | 99.8 |
| eyebrow | 50.4 | 70.9 | 55.7 | 100.0 |
| lip | 50.0 | 78.2 | 66.2 | 100.0 |
| hair | 49.7 | 63.2 | 46.8 | 99.8 |
| Total | 50.3 | 71.7 | 57.5 | 99.9 |

(d) SeqDeepFake comp. [63] with *CLIP* matching

| | AUC | F1 | mAP | Recall |
|---|---|---|---|---|
| nose | 55.8 | 31.6 | 55.9 | 20.5 |
| eyebrows | 50.4 | 2.5 | 52.0 | 1.2 |
| eyes | 54.0 | 37.3 | 53.7 | 27.8 |
| mouth | 52.7 | 20.5 | 53.7 | 12.4 |
| faceswap | 66.0 | 64.4 | 62.5 | 60.3 |
| Total | 55.8 | 31.3 | 55.6 | 24.5 |

(e) R-splicer with *contains* matching

| | AUC | F1 | mAP | Recall |
|---|---|---|---|---|
| nose | 54.4 | 66.6 | 58.5 | 99.9 |
| eyebrows | 51.7 | 66.6 | 52.7 | 99.9 |
| eyes | 53.5 | 66.3 | 53.7 | 100.0 |
| mouth | 52.8 | 66.8 | 54.1 | 100.0 |
| faceswap | 55.1 | 66.4 | 57.1 | 100.0 |
| Total | 53.5 | 66.5 | 55.2 | 100.0 |

(f) R-splicer with *CLIP* matching

Table 12: InstructBLIP [15] performance with *contains* and *CLIP* matching in open-ended VQA

| | AUC | F1 | mAP | Recall |
|---|---|---|---|---|
| Bangs | 51.3 | 9.1 | 61.1 | 5.2 |
| Eyeglasses | 50.5 | 3.0 | 63.7 | 1.6 |
| Beard | 50.2 | 1.2 | 61.1 | 0.6 |
| Smiling | 50.0 | 1.2 | 58.3 | 0.6 |
| Young | 50.0 | 76.8 | 62.3 | 99.9 |
| Total | 50.4 | 18.3 | 61.3 | 21.6 |

(a) SeqDeepFake attributes [63] with *contains* matching

| | AUC | F1 | mAP | Recall |
|---|---|---|---|---|
| Bangs | 49.6 | 73.6 | 58.5 | 99.6 |
| Eyeglasses | 47.2 | 75.7 | 60.5 | 99.1 |
| Beard | 58.0 | 73.9 | 64.7 | 99.3 |
| Smiling | 49.2 | 72.0 | 56.0 | 99.5 |
| Young | 50.5 | 74.9 | 60.0 | 99.5 |
| Total | 50.9 | 74.0 | 59.9 | 99.4 |

(b) SeqDeepFake attributes [63] with *CLIP* matching

| | AUC | F1 | mAP | Recall |
|---|---|---|---|---|
| nose | 50.1 | 0.9 | 55.5 | 0.4 |
| eye | 50.0 | 0.1 | 66.4 | 0.1 |
| eyebrow | 50.0 | 0.0 | 57.7 | 0.0 |
| lip | 49.7 | 0.4 | 67.5 | 0.2 |
| hair | 50.4 | 18.8 | 49.0 | 16.4 |
| Total | 50.0 | 5.1 | 59.2 | 4.3 |

(c) SeqDeepFake comp. [63] with *contains* matching

| | AUC | F1 | mAP | Recall |
|---|---|---|---|---|
| nose | 49.2 | 69.0 | 52.0 | 99.9 |
| eye | 46.9 | 77.3 | 61.7 | 99.8 |
| eyebrow | 49.5 | 70.8 | 54.1 | 99.9 |
| lip | 49.5 | 78.2 | 63.6 | 99.9 |
| hair | 49.6 | 63.3 | 46.1 | 99.9 |
| Total | 48.9 | 71.7 | 55.5 | 99.9 |

(d) SeqDeepFake comp. [63] with *CLIP* matching

| | AUC | F1 | mAP | Recall |
|---|---|---|---|---|
| nose | 54.0 | 16.8 | 55.2 | 9.3 |
| eyebrows | 50.4 | 3.3 | 52.1 | 1.7 |
| eyes | 50.4 | 9.3 | 51.6 | 5.4 |
| mouth | 53.0 | 19.4 | 54.0 | 11.8 |
| faceswap | 53.5 | 68.7 | 53.4 | 96.4 |
| Total | 52.3 | 23.5 | 53.2 | 24.9 |

(e) R-splicer with *contains* matching

| | AUC | F1 | mAP | Recall |
|---|---|---|---|---|
| nose | 50.8 | 66.6 | 52.0 | 99.9 |
| eyebrows | 49.2 | 66.6 | 49.3 | 99.9 |
| eyes | 49.9 | 66.3 | 49.2 | 99.9 |
| mouth | 51.0 | 66.8 | 51.0 | 100.0 |
| faceswap | 41.3 | 66.4 | 45.3 | 100.0 |
| Total | 48.5 | 66.5 | 49.3 | 99.9 |

(f) R-splicer with *CLIP* matching

Table 13: InstructBLIP [15] with T5-xxl LLM [13] performance, with *contains* and *CLIP* matching in open-ended VQA

|  | AUC | F1 | mAP | Recall |
|---|---|---|---|---|
| Bangs | 54.4 | 27.4 | 62.9 | 16.7 |
| Eyeglasses | 60.6 | 65.6 | 69.1 | 59.7 |
| Beard | 51.0 | 6.3 | 61.6 | 3.3 |
| Smiling | 51.9 | 11.5 | 59.5 | 6.2 |
| Young | 50.0 | 76.8 | 62.3 | 100.0 |
| Total | 53.6 | 37.5 | 63.1 | 37.2 |

(a) SeqDeepFake attributes [63] with *contains* matching

|  | AUC | F1 | mAP | Recall |
|---|---|---|---|---|
| Bangs | 45.8 | 46.3 | 56.2 | 40.7 |
| Eyeglasses | 56.1 | 65.3 | 65.7 | 65.9 |
| Beard | 55.0 | 60.3 | 63.4 | 59.2 |
| Smiling | 46.3 | 55.0 | 54.7 | 56.0 |
| Young | 50.2 | 50.6 | 61.9 | 43.7 |
| Total | 50.7 | 55.5 | 60.4 | 53.1 |

(b) SeqDeepFake attributes [63] with *CLIP* matching

|  | AUC | F1 | mAP | Recall |
|---|---|---|---|---|
| nose | 50.1 | 0.9 | 55.5 | 0.4 |
| eye | 50.0 | 0.1 | 66.4 | 0.1 |
| eyebrow | 50.0 | 0.0 | 57.7 | 0.0 |
| lip | 49.7 | 0.4 | 67.5 | 0.2 |
| hair | 50.4 | 18.8 | 49.0 | 16.4 |
| Total | 50.0 | 5.1 | 59.2 | 4.3 |

(c) SeqDeepFake comp. [63] with *contains* matching

|  | AUC | F1 | mAP | Recall |
|---|---|---|---|---|
| nose | 54.8 | 56.2 | 56.7 | 56.6 |
| eye | 58.2 | 57.7 | 68.5 | 49.4 |
| eyebrow | 51.2 | 56.1 | 55.1 | 56.8 |
| lip | 60.1 | 71.9 | 70.7 | 76.4 |
| hair | 54.1 | 57.3 | 48.3 | 70.4 |
| Total | 55.6 | 59.8 | 59.9 | 61.9 |

(d) SeqDeepFake comp. [63] with *CLIP* matching

|  | AUC | F1 | mAP | Recall |
|---|---|---|---|---|
| nose | 53.1 | 18.4 | 54.0 | 10.6 |
| eyebrows | 50.5 | 7.5 | 52.1 | 4.0 |
| eyes | 52.1 | 38.5 | 52.5 | 29.9 |
| mouth | 53.7 | 29.4 | 54.2 | 19.2 |
| faceswap | 59.5 | 61.9 | 57.1 | 63.7 |
| Total | 53.8 | 31.1 | 54.0 | 25.5 |

(e) R-splicer with *contains* matching

|  | AUC | F1 | mAP | Recall |
|---|---|---|---|---|
| nose | 53.5 | 57.2 | 53.5 | 63.7 |
| eyebrows | 52.7 | 59.1 | 51.9 | 69.6 |
| eyes | 51.4 | 57.2 | 50.7 | 66.0 |
| mouth | 54.3 | 62.2 | 53.7 | 78.3 |
| faceswap | 58.0 | 65.7 | 55.5 | 90.1 |
| Total | 54.0 | 60.3 | 53.1 | 73.5 |

(f) R-splicer with *CLIP* matching

Table 14: LlaVa-1.5 [43] performance, with *contains* and *CLIP* matching in open-ended VQA

|  | AUC | F1 | mAP | Recall |
|---|---|---|---|---|
| Bangs | 48.9 | 56.9 | 59.9 | 55.9 |
| Eyeglasses | 53.6 | 39.0 | 65.2 | 27.4 |
| Beard | 52.5 | 16.7 | 62.5 | 9.5 |
| Smiling | 51.1 | 12.1 | 59.0 | 7.1 |
| Young | 49.3 | 75.3 | 62.0 | 96.1 |
| Total | 51.1 | 40.0 | 61.7 | 39.2 |

(a) SeqDeepFake attributes [63] with *contains* matching

|  | AUC | F1 | mAP | Recall |
|---|---|---|---|---|
| Bangs | 46.4 | 73.5 | 56.9 | 99.9 |
| Eyeglasses | 51.9 | 75.9 | 65.9 | 100.0 |
| Beard | 57.9 | 74.1 | 64.0 | 100.0 |
| Smiling | 52.7 | 72.0 | 59.2 | 100.0 |
| Young | 47.5 | 75.1 | 58.8 | 100.0 |
| Total | 51.3 | 74.1 | 61.0 | 100.0 |

(b) SeqDeepFake attributes [63] with *CLIP* matching

|  | AUC | F1 | mAP | Recall |
|---|---|---|---|---|
| nose | 48.8 | 19.7 | 54.9 | 12.7 |
| eye | 49.7 | 3.2 | 66.4 | 1.6 |
| eyebrow | 49.9 | 7.1 | 57.7 | 3.8 |
| lip | 50.3 | 12.3 | 67.8 | 6.8 |
| hair | 49.3 | 43.2 | 48.4 | 43.9 |
| Total | 49.6 | 17.1 | 59.0 | 13.8 |

(c) SeqDeepFake comp. [63] with *contains* matching

|  | AUC | F1 | mAP | Recall |
|---|---|---|---|---|
| nose | 51.9 | 68.9 | 53.6 | 99.9 |
| eye | 50.0 | 77.4 | 63.1 | 100.0 |
| eyebrow | 48.7 | 70.8 | 54.1 | 100.0 |
| lip | 50.8 | 78.2 | 65.3 | 100.0 |
| hair | 46.8 | 63.3 | 44.5 | 100.0 |
| Total | 49.6 | 71.7 | 56.1 | 100.0 |

(d) SeqDeepFake comp. [63] with *CLIP* matching

|  | AUC | F1 | mAP | Recall |
|---|---|---|---|---|
| nose | 57.6 | 34.8 | 57.4 | 22.5 |
| eyebrows | 53.5 | 19.5 | 54.3 | 11.2 |
| eyes | 57.4 | 43.2 | 56.2 | 33.3 |
| mouth | 55.9 | 36.7 | 55.8 | 25.6 |
| faceswap | 68.8 | 73.8 | 63.6 | 84.6 |
| Total | 58.7 | 41.6 | 57.4 | 35.5 |

(e) R-splicer with *contains* matching

|  | AUC | F1 | mAP | Recall |
|---|---|---|---|---|
| nose | 56.8 | 66.6 | 60.0 | 100.0 |
| eyebrows | 54.5 | 66.6 | 56.2 | 100.0 |
| eyes | 56.5 | 66.3 | 56.0 | 100.0 |
| mouth | 55.0 | 66.8 | 55.6 | 100.0 |
| faceswap | 60.7 | 66.4 | 59.3 | 100.0 |
| Total | 56.7 | 66.5 | 57.4 | 100.0 |

(f) R-splicer with *CLIP* matching

