# OpenReview forum: "A Hitchhiker's Guide to Fine-Grained Face Forgery Detection Using Common Sense Reasoning"
_NeurIPS.cc/2024/Datasets_and_Benchmarks_Track — NeurIPS 2024 Track Datasets and Benchmarks Poster_

### Official Review · Reviewer_Gomc · 2024-07-11
**Comments on "A Hitchhikers Guide to Fine-Grained Face Forgery Detection Using Common Sense Reasoning"**

**Rating:** 7
**Confidence:** 4
**Correctness:** All the claims and evaluation design …
**Clarity:** This work is well-written and easy to…

**Review:**

- quality: the overall quality is relatively good. I believe that evaluating VLLMs' performance on deepfake datasets will provide valuable insights for future works. The writing style and paper structure is well-designed. The evaluation and analysis in both manuscript and supplementary seem to be comprehensive.
- clarity: the paper is well-written, with clear motivation, background introduction, and analysis. It is easy to follow from start to end, with engaging metaphors that enhance readability.
- originality and significance: this paper provides a new benchmarking dataset with high originality.

**Strengths:**

- the authors present an interesting work that uses zero-shot VLLMs to detect deepfakes. They employ four VLLMs and seven deepfake datasets to ensure a comprehensive evaluation.
- the authors propose three different types of questions as prompts，and study the impact of different synonyms.
- writing style: the paper is engaging and well-structured, with good logic and attractive metaphors.

**Additional Feedback:**

N/A

**Documentation:**

N/A

**Ethics:**

Not need.

**Limitations:**

- although the authors provide results on several datasets, I find that VLLMs achieve comparatively high performance only on SeqDeepFake and R-splicer, while failing on other more commonly used datasets. SeqDeepFake and R-splicer are not the most widely used datasets in the deepfake detection field. Moreover, I have some concerns about the bias problem in R-splicer, which uses different augmentations to generate pseudo deepfakes. The generated images contain some special artifacts, such as blending traces, which make the VLLM easier to detect. It cannot reflect the VLLMs’ performance on the “real” deepfake datasets.
- “Common Sense Reasoning” may not be suitable for this paper. The diversity of the answers from VLLMs is limited. It still belongs to binary classification or multi-label classification without in-depth analysis of the decision, especially since the reliability cannot be guaranteed.
- It is unclear to me what the black and purple lines in figure 2 represent. Why not use bars instead?

**Opportunities For Improvement:**

If the authors can address my concerns in the limitations, I would like to improve the socre.

**Relation To Prior Work:**

As far as I know, little work focused on zero-shot VLLMs has been explored in the deepfake detection field before 2024.

**Summary And Contributions:**

This paper leverages several Vision and Large Language Models (VLLMs) to build a new benchmark for fine-grained deepfake detection. The authors approach the deepfake detection problem as a Visual Question Answering (VQA) task and develop three evaluation protocols: binary classification and multi-label classification with/without fixed choices. They also explore new matching strategies, such as CLIP distance, which I find interesting. As far as I know, little work focused on zero-shot VLLMs has been explored in the deepfake detection field before 2024. Based on the quality and contribution, I believe this work deserves a positive score.

---

> ### Author Rebuttal · Authors · 2024-08-16
>
> 1. **VLLM performance**: We thank the reviewer for the feedback. The reviewer is correct in noting that VLLMs perform better on SeqDeepFake and R-Splicer and that these datasets may not fully represent real-world deepfake scenarios. This reflects a broader limitation in the field: the scarcity of real-world datasets with fine-grained annotations affects task-specific training and evaluation of VLLMs. We will expand on this limitation in our revised manuscript, as also discussed with Reviewer a29e. In addition, at Reviewer's t8Gi suggestion, we include qualitative analysis of the VLLMs included in the study to the adjacent task of image editing detection in a very challenging dataset in the Appendix. Our study highlights this critical gap and aims to inform the community about the need for more comprehensive real-world data to improve VLLM performance on diverse deepfake detection tasks.
>
> 2. **''Common Sense Reasoning''**: In the context of VLLMs, common sense reasoning refers to a model’s ability to apply general world knowledge and everyday understanding to generate responses enabled by the very large vision-language pertaining to the models. In the context of the paper, this is demonstrated by the ability of the models to make zero-shot predictions with no prior training on face forgery detection and justify this response --the latter shown more clearly in the open-ended VQA evaluation, as the responses are longer. We will clarify this in the introduction, where common sense reasoning is first introduced as a term.
>
> 3. **Fig.2**: We welcome the reviewer's suggestion and will replace the lines with bars for consistency in the camera-ready.

---

> > ### Comment · Reviewer_Gomc · 2024-08-30
> >
> > Thank you for response my concerns, I prefer to maintain my original score.

---

### Official Review · Reviewer_a29e · 2024-07-23
**Comments of the manuscript A Hitchhikers Guide to Fine-Grained Face Forgery Detection Using Common Sense Reasoning**

**Rating:** 7
**Confidence:** 4
**Correctness:** I do not see any correctness problems…
**Clarity:** Yes.

**Review:**

Please see Strengths and Limitations.

**Strengths:**

1. The paper is well organized and easy to follow.

2. The idea of using VLLMs to perform deepfake detection is novel.

3. The experiments are conprehensive.

**Additional Feedback:**

Please see Opportunities For Improvement.

**Documentation:**

Yes

**Ethics:**

No ethical concerns found.

**Limitations:**

1. Explainability refers to the reasons why the detection model makes its decisions. Even though existing VLLMs can provide more fine-grained deepfake detection, the rationales behind the model's decisions remain ambiguous.

2. Compared to the SOTA deepfake detection methods, existing VLLMs exhibit limited detection performance. While explainability is important, detection accuracy should be the first priority in the task of deepfake detection.

3. How do the authors obtain the labels for fine-grained deepfake detection in the DFDC, CelebDF, and WildDF datasets? As far as I know, these datasets do not indicate the manipulated facial regions.

4. This work designs an EM metric that considers the response correct if the class name is contained in the response. Given that responses are usually long and detailed, what if the responses exhaustively contain all classes? This case would result in a high EM score and a high false positive rate.

**Opportunities For Improvement:**

1. The experiment section lacks powerful VLLMs, such as GPT-4 and Gemini. It would be interesting to examine the performance of these models on a subset of the used datasets.

2. It is interesting to explore the few-shot face forgery detection performance of existing VLLMs.

3. As the prediction is polarized, the usage of AUC is meaningless.

4. There is a lack of deep analysis on the detection performance. It is unclear what lessons can be learned from the experimental results

**Relation To Prior Work:**

Yes.

**Summary And Contributions:**

This work aims to enhance the explainability of face forgery detection by exploiting Vision and Large Language Models' (VLLM) outstanding reasoning capability. The established benchmark examines the capability of VLLM in binary and fine-grained deepfake detection in an VQA manner.

---

> ### Author Rebuttal · Authors · 2024-08-16
>
> 1. **Comparison with GPT4**: We thank the reviewer for the suggestion. In the camera-ready version of the paper, we include GPT4o binary performance on all datasets. As DFDC, R-splicer, DFW, and FF++ datasets are very large, we evaluate on a 5k sample sub-set for each. The results can be seen in the table below. We note that this was initially excluded due to concerns about fairness, as we cannot verify GPT4 training data. GPT4o vastly outperforms the other tested VLLMs on three datasets (DFDC, CelebDF and DFW) and is comparable to BLIP and LlaVa on the others tested, with the exception of FF++, where the performance is very poor. However, as for the larger datasets we evaluated on a subset, the results may be susceptible to some degree of bias from the sampling, which needs to be considered when comparing the models. All metrics and comparisons will be included in the camera-ready version of the main paper.
>
> 2. **Few-shot**: In the camera-ready version of the paper, we will include the performance of LlaVa-1.5 with its vision encoder fine-tuned contrastively on FF++. The architecture with task-specific vision encoder performs consistently better across all datasets in terms of F1, with +7\% improvement within the dataset and +1-3\% improvement across datasets, thus showing good cross-dataset performance even without updating the weights of the LLM. We note that end-to-end training of the models, including the LLM, would require a method for caption generation so as to avoid catastrophic forgetting, which is outside the scope of the proposed framework. However, as it is a very important observation, we include this key point in the camera-ready paper's limitations and future work section.
>
> 3. **Metrics used**: We agree with the comments regarding AUC. The predictions are indeed polarised, and thus, AUC is not informative. We discussed this briefly in the main paper (line 210), where we clarified that we include it for reference. In the camera-ready version of the paper, we further discuss the evaluation metric's usefulness in the results section to inform future works.
>
> 4. **Analysis of results**: Thank you for your feedback. We acknowledge that our analysis has been primarily technical, with some key findings included in the "Overall Performance" and "Limitations and Future Work" subsections of Section 5. As we agree on the importance of a more in-depth analysis of how VLLMs currently perform and can benefit face forgery detection, we will expand the limitations and future work section. Specifically, our results indicate a need for task-specific models, as demonstrated by the comparison of the zero-shot performance of these VLLMs with previous works. Additionally, we identify a gap in available datasets; there is a lack of datasets where fine-grained manipulation areas or artefacts are described in natural language, which limits the ability to train VLLMs end-to-end. Addressing this gap will require either transforming categorical labels into captions or creating human-annotated datasets.
>
> 5. **Explainability**: Model explainability does indeed often involve understanding how a model arrives at its decisions, typically through post-hoc analysis like attention maps and Grad-CAM heatmaps. For language generation models, explainability can be demonstrated by generating outputs directly conveying reasoning (e.g., ''eyes are blurry'') or presenting information in a human-readable format. In the introduction, we will further clarify this aspect of inherent explainability in VLLMs.
>
>
> 6. **Comparison with previous SoTA**: Thank you for highlighting this important consideration. Our work proposes a systematic and fair evaluation framework for VLLMs, which is crucial for assessing their potential and limitations in the context of deepfake detection. By providing this framework, we offer key insights into how VLLMs perform and how they can be integrated into the field of deepfake detection. This framework is extendable, allowing for future improvements and fair comparisons of methods. Our goal is to advance the understanding of VLLMs in this domain, enabling the community to identify opportunities for enhancement and refine approaches based on a standardised evaluation protocol. In summary, while VLLMs may not yet match SOTA methods in detection accuracy as there is a lack of task-specific models, our framework contributes to the field by offering a structured way to evaluate and develop these models further, which is essential for progress.
>
> 7. **Fine-grained labels**: We clarify that the fine-grained comparison is performed for the datasets that contain fine-grained labels, namely SeqDeepFake and R-splicer. The other datasets mentioned are used to evaluate the models on the binary task. This is discussed in sections 4.2 and 5.1.
>
> 8. **''Contains'' and false positive rate**: Thank you for pointing out this important consideration. We acknowledge that the contains metric, which matches a response to a discrete class if it contains the class name, may indeed lead to false positives when responses are lengthy and include multiple class names. Our results section already identifies this issue in the context of multiple-choice VQA tasks (line 264), where the prompt may introduce some degree of bias to the model. However, we have not observed this problem with open-ended questions in our experiments, which can also be seen in the qualitative analysis. We will elaborate on this as a potential limitation of the matching strategies in the camera-ready version of the paper.
>
> |                   | GPT4  |
> |-------------------|-------|
> |  SeqDeepFake Attr | 97.11 |
> | SeqDeepFake Comp. | 97.26 |
> |     R-splicer     | 97.67 |
> |        FF++       | 22.19 |
> |        DFDC       | 99.16 |
> |      CelebDF      | 65.34 |
> |        DFW        | 99.14 |
> |     StyleGAN2     | 67.31 |
> |     StyleGAN3     | 67.99 |
>
> Table 1. GPT4o binary classification performance on nine benchmark datasets in terms of F1

---

> > ### Comment · Reviewer_a29e · 2024-08-30
> >
> > Thanks for the authors detailed response and the additional experiments. I have enhanced the rating to 7.

---

### Official Review · Reviewer_t8Gi · 2024-07-24
**A valuable addition to deepfake detection research**

**Rating:** 5
**Confidence:** 5
**Correctness:** Yes
**Clarity:** Yes

**Review:**

Pros:

- Extensive comparisons are conducted on several datasets, providing a detailed analysis of the performance of various VLLMs.

- The structure of the paper is well-organized, with clear sections outlining the methodology, experiments, and results, making it easy to follow.

- Diagrams and tables are used effectively to illustrate the evaluation framework and present the results.

- This is one of the first studies to systematically evaluate VLLMs in the context of fine-grained deepfake detection, filling a gap in the current research.

Cons:

-  The paper could benefit from a more detailed description of the datasets used, particularly concerning the annotation process. A comprehensive explanation of how annotations were built for existing datasets and how they were used to train the VLLMs would add clarity.

- The paper includes a significant amount of technical terminology related to VLLMs, which might be challenging for readers who are not deeply familiar with the field.

- There are several concurrent related works, which to some extent, undermine the novelty of this paper's contributions. For instance:

Chang, You-Ming, Chen Yeh, Wei-Chen Chiu, and Ning Yu. “AntifakePrompt: Prompt-Tuned Vision-Language Models Are Fake Image Detectors.” arXiv, November 2, 2023. http://arxiv.org/abs/2310.17419.

Wang, Yabin, et al. "Linguistic Profiling of Deepfakes: An Open Database for Next-Generation Deepfake Detection." arXiv preprint arXiv:2401.02335 (2024).

Keita, Mamadou, Wassim Hamidouche, Hassen Bougueffa, Abdenour Hadid, and Abdelmalik Taleb-Ahmed. “Harnessing the Power of Large Vision Language Models for Synthetic Image Detection.” arXiv, April 3, 2024. http://arxiv.org/abs/2404.02726.

- The paper lacks detailed information on how metrics like AP (Average Precision) and AUC (Area Under the Curve) are calculated. Specifically, it is unclear how the binary classification prediction scores are obtained from the LLM outputs.

- Most datasets used involve entire face replacements, which might not fully test the explainability aspect intended by the paper. Since the entire face is replaced, and other parts like ears, neck, and body remain unchanged, this homogeneity does not adequately showcase the LLMs' reasoning capabilities. Using a different type of dataset, such as those focused on image editing detection, might better demonstrate the reasoning abilities of LLMs.

**Strengths:**

See Review

**Additional Feedback:**

NAN

**Documentation:**

NO

**Limitations:**

See Review

**Opportunities For Improvement:**

See Review

**Relation To Prior Work:**

Not clear enough

**Summary And Contributions:**

The paper innovatively transforms face forgery detection tasks into Visual Question Answering (VQA) multi-label problems, systematically evaluating the performance of Vision Large Language Models (VLLMs) in fine-grained detection scenarios. By introducing a unified evaluation framework, the study conducts extensive comparisons across multiple datasets, highlighting both the strengths and limitations of current VLLMs in the context of deepfake detection. The findings reveal that while VLLMs offer significant potential in terms of explainability and intrinsic reasoning, they may not yet surpass the performance of specialized models designed specifically for deepfake detection tasks.

---

> ### Author Rebuttal · Authors · 2024-08-16
>
> 1. **Descriptions of datasets**: We appreciate this suggestion and will expand both the main text and the supplementary sections to include more comprehensive details on the datasets, particularly focusing on the annotation process. Specifically, Section 4.2 contains a description of the datasets used for evaluation and the supplementary material that offers a more detailed explanation of the datasets employed to train the VLLMs evaluated in this work.
>
>
> 2. **VLLM terminology**: For clarity, the camera-ready version of the paper will include a section in the Appendix covering preliminaries on VLLMs and VQA tasks, specifically regarding conditional language generation.
>
> 3. **Concurrent works**: As foundation models continue to gain traction across various tasks, it is natural to see an increasing number of works emerging in the field of face forgery detection as well. We respectfully disagree with the reviewer’s concern that the novelty of our proposed method is undermined by the listed concurrent works. Specifically, the works by Chang et al. (2023), which we reference in our related work section, and Keita et al. (2024) focus on fine-tuning CLIP-like models and evaluating them on the binary classification task within a retrieval setting. Similarly, Wang et al. (2024) introduce a new dataset of generated images, not limited to face forgery. Our work takes a novel approach and addresses challenges in deepfake detection using language generation, which distinguishes it from the aforementioned pre-prints. Namely, our main contributions are transforming the discriminative task to a VQA problem for both binary and fine-grained labels, establishing a framework to systematically and consistently evaluate VLLMs and providing insights to the community from analysis of current language generation methods. We will include the listed works in the related works section and further clarify these distinctions.
>
> 4. **Clarification on metrics**: Since VLLMs generate responses in natural language, we need a method to map these to discrete classes. To achieve this, we propose three matching strategies: exact match, contains, and CLIP distance. For the binary task, we use exact match, classifying a sample as fake if the model responds ``Yes'' to the prompt, i.e. assigns $1$. For fine-grained tasks, we use the contains strategy, which labels a class as positive if its name is present in the response, and CLIP distance, which measures the cosine similarity between the text embeddings of the response and each class name, labelling all classes above a threshold as positive. These predictions are then compared to the ground truth as usual. We refer the reviewer to Section 3.3 for a detailed discussion on matching generated language to class names to obtain predictions.
>
> 5. **Explainability capabilities**: We thank the reviewer for the suggestion; we perform experiments on the very challenging MagicBrush dataset [1] to assess response quality. As image editing datasets are typically language-driven, they do not contain discrete classes; therefore, even though akin to deepfake detection, the task does not allow for all evaluation protocols proposed in this work to be used. We can, however, compare the responses of the tested VLLMs to the prompts used to generate the edited images and compute their Bert score, which is part of the qualitative analysis of our work. We will include the evaluation below in the supplementary material of our camera-ready paper. Furthermore, we would like to highlight that in the samples of generated responses included in the supplementary material (Fig. 7), we see that even when the entire face of the subject is manipulated, there is rationale in the response (''appears retouched'', ''distorted expression''), so we think that common sense reasoning still has a significant contribution even in cases of face swaps.
>
> |                  | Precision | Recall | F1    |
> |------------------|-----------|--------|-------|
> | Blip             | 85.44     | 84.37  | 84.57 |
> | InstructBlip     | 84.56     | 85.09  | 84.82 |
> | InstructBlip-xxl | 81.07     | 84.55  | 81.88 |
> | LlaVa~1.5        | 83.08     | 84.77  | 83.87 |
> Table 1: Bert score of selected VLLMs on MagicBrush dataset
>
> [1] Zhang, K., Mo, L., Chen, W., Sun, H. and Su, Y., 2024. Magicbrush: A manually annotated dataset for instruction-guided image editing. Advances in Neural Information Processing Systems, 36.

---

### Official Review · Reviewer_BdnU · 2024-07-26
**Review on Paper #98**

**Rating:** 6
**Confidence:** 4
**Correctness:** N.A
**Clarity:** Yes

**Review:**

The idea of using LVLM is not surprising as it has been widely used for general vision tasks. But the exploration of using LVLM for deepfake is still worth trying. The writing is clear and well-organized. The evaluation and analysis are comprehensive.

**Strengths:**

1. It is novel to use LVLM for deepfake by treating this as a VQA problem. However, it is still needed to emphasize why treating this as VQA is better.
2. The writing is good and easy to read.

**Additional Feedback:**

Please check the Limitation.

**Documentation:**

N.A

**Ethics:**

N.A

**Limitations:**

1. Lack of enough comparison  In the experiments of SeqDeepfake and R-Splicer, the performances are significant. But there are other commonly used dataset, such as DFFD, FaceForensic++, DFDC, etc. Should make necessary comparison.
2.  Despite good performance, why use of common sense reasoning is still not well explained. Please give insights on why it is better.  It is not conclusive that Common Sense Reasoning is better for Deepfake. Is there any limitation?

**Opportunities For Improvement:**

Improve the analysis of experiments. Please check the Limitation.
Supplement detailed analysis of why using Common Sense Reasoning

**Relation To Prior Work:**

1. Lack of enough comparison  In the experiments of SeqDeepfake and R-Splicer, the performances are significant. But there are other commonly used datasets, such as DFFD, FaceForensic++, DFDC, etc. Should make the necessary comparison.

**Summary And Contributions:**

This paper utilizes a Large Vision and Language Model (LVLM) for deepfake detection by formulating the deepfake detection as a VQA task. Three evaluation protocols are proposed

---

> ### Author Rebuttal · Authors · 2024-08-16
>
> 1. **Comparisons on other datasets**: We are pleased to see the reviewer recognises the significant performance on SeqDeepFake and R-Splicer. As the reviewer correctly points out, other commonly used datasets, such as FF++ and DFDC, are relevant to this task. These datasets are indeed included in our evaluation for the binary classification task. Specifically, in Section 3.1, we discuss our methodology for binary classification using VLLMs. In Section 4.2, we detail the datasets utilised in the evaluation, and in Section 5.1, we present results across all datasets for the binary classification task. However, as only SeqDeepFake and R-Splicer provide fine-grained labels, quantitative comparison for the fine-grained classification task is only possible with these datasets.
>
> 2. **VQA justification**: The primary motivation stems from the explainability capabilities VLLMs provide, which is crucial for understanding and trusting model decisions, especially in sensitive tasks like deepfake detection. As the output of VLLMs is generated language instead of discrete categories, the output allows for a direct explanation of each decision, which can be seen in the qualitative analysis of our study and the samples shown. Additionally, the very large pretraining of VLLMs enables the incorporation of extensive knowledge beyond the specific task, which shows promise in enhancing model performance by leveraging external, real-world understanding. These motivations are discussed in more detail in the introduction of our paper, and we will make an effort to expand on these points further, to ensure clarity. Furthermore, we acknowledge the current limitation of lacking task-specific models and appropriate datasets for end-to-end training, which is a key finding of our study and is mentioned in the "Limitations and Future Work" in Section 5. We will update the limitations section to further acknowledge and inform the community of the need for more specialized models and datasets.

---

### Author Rebuttal · Authors · 2024-08-16

We thank the reviewers for their thoughtful feedback and are pleased that they recognise the strengths of our proposed method. We appreciate all reviewers' positive comments on the "novel use of VLLM for deepfake" (Bdnu) and "innovative transformation of face forgery detection tasks into Visual Question Answering (VQA) multi-label problems" (t8Gi). We are glad that all reviewers found our method to be clear, well-written, and supported by "comprehensive experiments"(a29e), which is crucial in "filling a gap in the current research" (t8Gi).  We are also encouraged that the reviewers recognize "significant potential in terms of explainability and intrinsic reasoning" (t8Gi) and see this work "provides valuable insights for future works"(Gomc). Finally, we are very pleased all reviewers acknowledge the writing quality as "well-written, with clear motivation, background introduction, and analysis" (Gomc).

Furthermore, we would like to clarify some general comments.
1. **Explainability and Common Sense Reasoning**: One of the key motivations for using VLLMs in deepfake detection is the inherent explainability of natural language, that is, the ability of language generation models to directly convey reasoning for each prediction in a format where non-experts can easily understand. Furthermore, as VLLMs show common sense reasoning capabilities --the ability to make inferences about the world that go beyond the explicit information-- harnessing external knowledge in face forgery detection may provide significant advantages. These aspects of language generation models are more clearly demonstrated in the qualitative analysis, sample responses in the Appendix, and experiments on the adjacent task of image editing where tested models show a high Bert score. The aim of this work, and a key contribution, is to propose a unified framework that allows for a fair and systematic comparison of VLLMs in face forgery detection by transforming the discriminative detection task to VQA.
2. **Fine-grained label availability**: The majority of available datasets and previous works address face forgery as a binary classification, where an image is either fake or real, a coarse categorisation that is not always adequate. The majority of current works in face forgery detection rely on pseudo-fake datasets for fine-grained artefacts, as covered in the Related work section of our paper. However, identifying fine-grained areas of manipulation is an important element in assessing the explainability and reasoning of models beyond a binary classification. The proposed framework introduces three protocols to assess VLLM capabilities in both binary and fine-grained detection. For the former, we provide an extensive evaluation of several VLLMs in nine popular benchmarking datasets. For the latter, current comparisons are conducted on SeqDeepFake and R-splicer datasets, but are limited by the availability of data, which is a key finding of this study.
3. **Comparison with previous SoTA**: The aforementioned lack of fine-grained data and limited use of language generation models in deepfake detection results in a lack of task-specific models. As such, the comparison with previous SoTA shows that even though the tested VLLMs achieve good zero-shot performance and hold promise in advancing the field, their performance is still behind that of discriminative task-specific approaches, which is another key finding of the study. However, experiments with a fine-tuned vision encoder show improved performance even without updating the language model's weights, a finding that is crucial in informing future works.

---

### Author Response · Authors · 2024-09-02
**Discussion Summary**

We thank the Area Chair (AC) for overseeing the review process and to the reviewers for their constructive feedback and active engagement during the discussion. In this work we propose a novel approach to deepfake detection by reframing the task as a Visual Question Answering (VQA) multi-label problem. We introduce a systematic and unified evaluation framework that enables consistent assessment across various Vision-Language Large Models (VLLMs). Our comprehensive analysis and comparison of these models, including an ensemble approach and a fine-tuned architecture, offer new insights into their strengths and limitations in deepfake detection.

The primary concerns raised by the reviewers focused on the motivation for explainability, the role of common sense reasoning, the availability of fine-grained labels, and comparisons with previous state-of-the-art (SoTA) methods. We addressed these concerns by clarifying that explainability in our context refers to the ability of VLLMs to directly convey reasoning for each prediction in a format where non-experts can easily understand. We highlighted that VLLMs possess common sense reasoning capabilities, that is the ability to leverage external knowledge in face forgery detection. We also discussed the scarcity of datasets with fine-grained annotations, which limits the training and evaluation of VLLMs in this domain, and we committed to expanding on this in the limitations of the camera-ready version. Additionally, we conducted further experiments, including the use of a fine-tuned vision encoder, ChatGPT-4, and qualitative evaluation on a challenging image editing dataset, to strengthen our comparison with previous SoTA.

Following our rebuttal, Reviewer-a29e acknowledged "our detailed response and additional experiments", raising their score to 7. Reviewer-Gomc appreciated our efforts to address their concerns and maintained their already favorable score of 7. Reviewers BdnU and t8Gi have yet to respond, but our rebuttals comprehensively address their concerns. Specifically:

* Reviewer BdnU: We recognize the favorable score of 6 and plan to acknowledge additional limitations in the camera-ready version.
* Reviewer t8Gi: We clarified the key differences between our work and concurrent studies on VLLMs, noting that some are already discussed in our Related Work section. We propose to expand this section to further highlight the differences with the listed works. Furthermore, we have committed to include experiment on the task of image editing in the Appendix of the camera-ready paper.

We will integrate all the reviewers' suggestions and clarifications into the final version of our paper. We sincerely thank the reviewers for their contributions to improving our work.

Sincerely,

Submission #98 Authors

---

### Decision · Program_Chairs · 2024-09-26

**Decision:**

Accept (Poster)

**Comment:**

Thank you for the hard work of the reviewers and the detailed responses of the authors. Overall, this work has been recognized by the reviewers. The reviewers pointed out areas where the discussion needs to be improved, and the authors also promised to improve. Therefore, I recommend accepting this paper.